# Amino acid variability, tradeoffs and optimality in human diet

Ziwei Dai [1,2], Weiyan Zheng[2] & Jason W. Locasale [1] ✉

Studies at the molecular level demonstrate that dietary amino acid intake produces substantial effects on health and disease by modulating metabolism. However, how these effects may manifest in human food consumption and dietary patterns is unknown. Here, we develop a series of algorithms to map, characterize and model the landscape of amino acid content in human food, dietary patterns, and individual consumption including relations to health status, covering over 2,000 foods, ten dietary patterns, and over 30,000 dietary profiles. We find that the type of amino acids contained in foods and human consumption is highly dynamic with variability far exceeding that of fat and carbohydrate. Some amino acids positively associate with conditions such as obesity while others contained in the same food negatively link to disease. Using linear programming and machine learning, we show that these health trade-offs can be accounted for to satisfy biochemical constraints in food and human eating patterns to construct a Pareto front in dietary practice, a means of achieving optimality in the face of trade-offs that are commonly considered in economic and evolutionary theories. Thus this study may enable the design of human protein quality intake guidelines based on a quantitative framework.

Diet is generally considered to be a major determinant of human health and disease[1–5]. Numerous dietary recommendations, such as the Dietary Guidelines for Americans[6], have been developed. These dietary recommendations often focus on two major goals: to increase the diversity and nutrient density of the foods consumed, and to reduce the intake of certain components known to increase risk of disease[7–9]. Such restrictions involve limiting the intake of certain types of carbohydrate and fat such as added sugar, saturated fat and trans-fat, and has rationale based on epidemiology, human[10–12] and model organism research[13,14]. While it has been widely acknowledged that the types of dietary carbohydrate and fat are important determinants of the quality of a diet, protein the other macronutrient[15], is often neglected. In most human nutritional studies albeit with exceptions, protein is considered as a single variable and often held constant[16]. Nevertheless, each amino acid has its specific metabolism[17] and is important for numerous cellular and physiological processes. A growing number of studies shows that variation in dietary intake of amino acids such as serine, glycine, asparagine, histidine, and methionine mediates health and disease

including cancer through defined molecular mechanisms[18–28]. Altogether there is a rationale for investigating in a systematic manner amino acid intake in human diets and possible consequences on health.

In this study, we investigated the variability of amino acids in human food and diets and find variability commensurate with what is observed in fats and carbohydrates. Based on optimizing associations with health status, we use these analyses to devise guidelines for dietary amino acids. Finally, we implement machine learning algorithms to design personalized diets based on amino acid intake that correspond to optimality in specified health statuses.

## Results

### Amino acid landscape of human food

To characterize the variability of amino acid levels in human food, we first constructed a database consisting of amino acid profiles in three levels of human dietary components: individual foods, dietary patterns or representations of patterns of food consumption (e.g. Western,

[1]Department of Pharmacology and Cancer Biology, Duke University School of Medicine, Durham, NC 27710, USA. [2]Department of Biology, School of Life Sciences, Southern University of Science and Technology, Shenzhen 518055, China. ✉e-mail: dr.jason.locasale@gmail.com

Mediterranean, Japanese, Keto, etc), and dietary profiles containing daily reported food intake (Fig. 1). The abundance of 18 amino acids in 2,335 foods was collected based on nutritional profiles in the United States of America Department of Agriculture National Nutrient Database for Standard Reference Legacy Release (USDA SR) (Fig. 1a, Methods). 18 of the 20 amino acids were considered because during quantitation, amino acids which largely exist in protein-bound forms, require hydrolysis into free amino acids during which amino groups from glutamine and asparagine are also hydrolyzed to make glutamic and aspartic acid. Thus, the abundance of glutamic acid and aspartic acid from measurements of free amino acid levels reflects the total abundance of glutamate and glutamine, and the total abundance of

aspartate and asparagine, respectively. The distributions of amino acid abundance over 2,335 foods show that each amino acid has considerable variability across foods (Coefficient of variation > 0.2 for all amino acids, Fig. 1b), and amino acids most abundant in human food are glutamine/glutamate (median = 0.16 g/g total amino acids), asparagine/aspartate (median = 0.095 g/g total amino acids), leucine (median = 0.082 g/g total amino acids), and lysine (median = 0.076 g/g total amino acids). On the other hand, amino acids with the lowest abundance in human foods are cystine (median = 0.012 g/g total amino acids), tryptophan (median = 0.012 g/g total amino acids), methionine (median = 0.024 g/g total amino acids), and histidine (median = 0.028 g/g total amino acids). This ordering largely resembles the

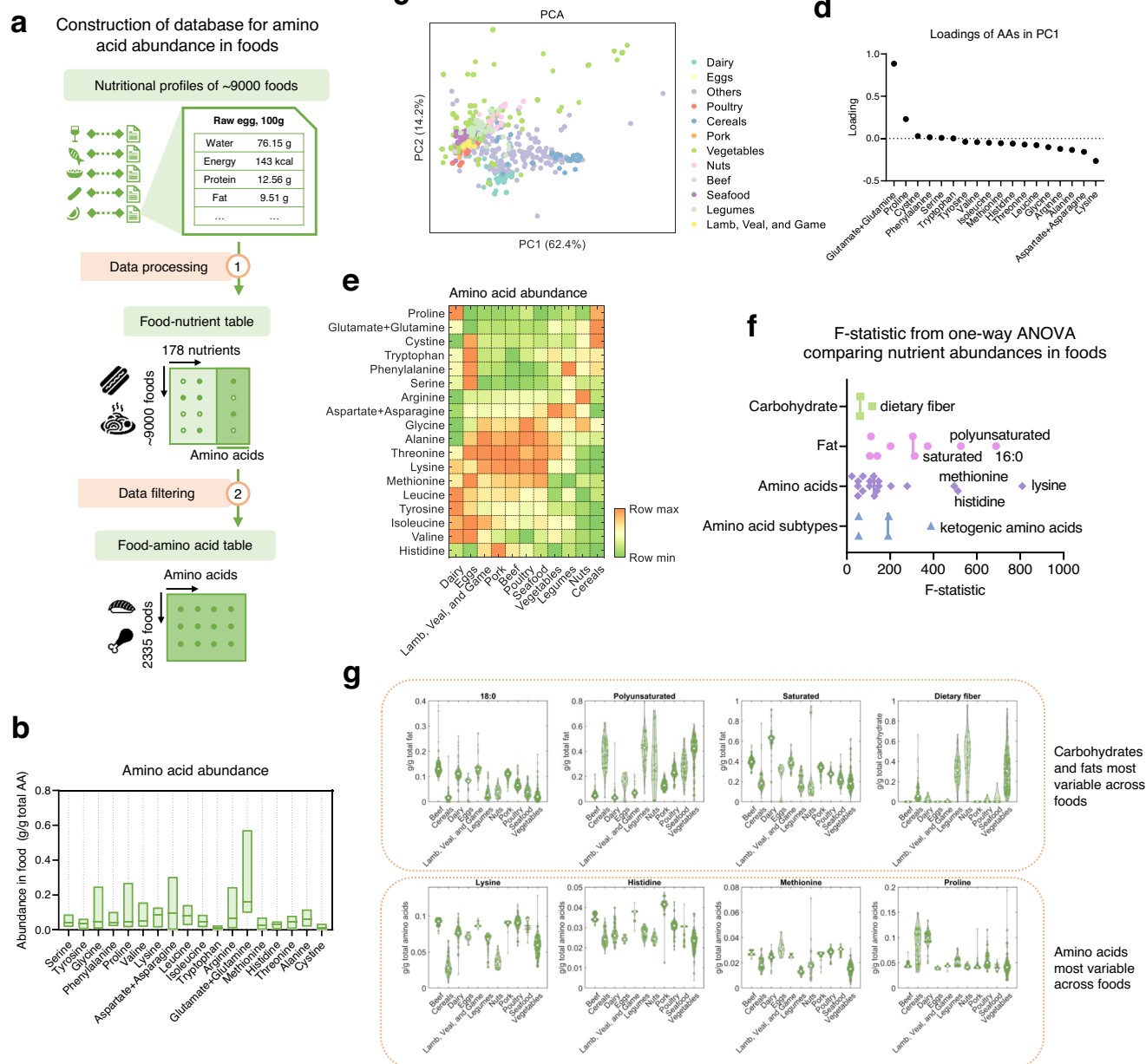

**Fig. 1 | Amino acid landscape of human foods. a** Workflow for construction of the database for amino acid abundances in human foods. Food images used with permission from Microsoft. **b** Ranges of amino acid abundance in human foods. The horizontal lines indicate median values. The upper and lower bounds of boxes indicate the range of data. n = 2335 foods. **c** Principal components analysis (PCA) of amino acid profiles in human foods. Each dot represents a food. Colors of the dots indicate different categories of the foods. **d** Loading of amino acids in the first principal component in the principal components analysis (PCA). **e** Average amino acid abundance in different categories of human foods. **f** F-statistic values from one-way analysis of variance (ANOVA) comparing abundance of single amino acids, different types of amino acid, different types of carbohydrate, and different types of fat across human foods. **g** Violin plots showing the distributions of abundance of amino acids, carbohydrates, and fats that are the most variable across human foods. The circles indicate median values. Green dots indicate individual values.

abundance of amino acids in the proteomes which are conserved across living organisms[29,30]. Principal component analysis (PCA) shows that amino acid abundances can be clustered by different categories of foods (Fig. 1c, Methods). Highly variable amino acids include those whose dietary modulation has molecular links to cancer progression and health outcomes, such as methionine (0.031 g/g total amino acids in eggs compared to 0.013 in legumes) and serine (0.076 g/g total amino acids in eggs compared to 0.039 in lamb, veal, and game meat). The loadings in the first principal component (Fig. 1d) indicate that the amino acids with the largest contribution to the first principal component are glutamate/glutamine, proline, lysine, and aspartate/asparagine. Each of these amino acids is enriched in at least one major category of human foods (Fig. 1e), suggesting that the first principal component mainly reflects differences in amino acid levels across categories of foods. To quantify the variability of amino acid abundance across foods, we computed the F-statistic from one-way analysis of variance (ANOVA), and compared the resulting F-statistic values with those of carbohydrates (i.e. dietary fiber and sugar) and fats (i.e. saturated fat, monounsaturated fat, and polyunsaturated fat). Notably, we found that the ANOVA F-statistics for both single amino acids and five subtypes of amino acids (essential amino acids, nonessential amino acids, branched-chain amino acids, ketogenic amino acids, and glucogenic amino acids) were comparable to or higher than those for carbohydrates and fats (Fig. 1f, Methods), especially for the amino acids methionine, histidine, lysine, and proline (F-statistic = 816.2 for methionine, 566.1 for histidine, 504.3 for lysine, and 362.9 for proline compared to the range of 45.0 to 119.6 for carbohydrates and the range of 125.2 to 746.3 for fats, Fig. 1f, g), highlighting the variability of amino acid abundance in foods which has been largely overlooked previously.

We also compared amino acid profiles of plant-based versus animal-based foods and computed the coefficient of variation (CV) as a metric quantifying the variability of amino acid profiles among these foods (Supplementary Fig. 1). We found that although the differences in mean amino acid levels between plant- and animal-based foods were subtle as previous studies show[31], there is high variability in amino acid profiles among plant- or animal-based foods (higher values of CV among either animal- or plant-based foods compared to the CV among all foods for all amino acids except for histidine, Supplementary Fig. 1b). This finding indicates that simply comparing the amino acid profiles between plant- and animal-based food without consider all other relevant variables underestimates greatly the variability of amino acids in human foods. Taken together, these results suggest that differences in food intake due to the high variability in amino acid content may lead to differences in physiological and cellular effects on metabolism.

## Human dietary patterns are variable in amino acid content

Dietary patterns can be grouped according to eating patterns that often have a cultural or societal element. They can be characterized by a combination of certain types of foods consumed (e.g. Mediterranean diet, which includes high amounts of plant-based foods, high to moderate amounts of seafood, low consumption of red meat, and olive oil as the main source of added fat[32]), or a specific intake profile of certain nutrients (e.g. ketogenic diet, which is defined by very high intake of fat and very low intake of carbohydrate). Adherence to certain dietary patterns, such as the Mediterranean diet or Japanese diet, has been associated with increased lifespan and lower risk of disease[33–35]. Moreover, some emerging dietary patterns, such as the ketogenic diet and the Paleo diet, have recently been shown in some settings to have benefits on metabolic health, neural function, and longevity[36–39]. However, it is unclear whether these dietary patterns differ in their amino acid content, and whether the variability in amino acid abundance across dietary patterns contributes to the health outcomes associated with these diets.

To further understand the relationship between human dietary patterns and amino acid intake, we next developed an algorithm to quantitatively evaluate amino acid abundance in ten representative human dietary patterns (Fig. 2a, Supplementary Fig. 2, Supplementary Methods). Among these dietary patterns, the Mediterranean diet and Japanese diet are two traditional diets believed to have beneficial influences on health, while the Dietary Approaches to Stop Hypertension (DASH) diet consists of consumption of a variety of low-fat and minimally processed foods, and the American diet, which represents the dietary behaviors of a typical individual in western society is also considered. We also include diets that restrict the consumption of certain foods (Paleo diet, vegetarian diet, plant-based diet), diets limiting carbohydrate intake (ketogenic diet, Atkins diet), and a USDA recommended diet defined based on the daily nutrient intake goals in the USDA 2015-2020 dietary guidelines for Americans[6]. We first computed the range of amino acid intake (i.e. grams of each amino acid consumed per day) for each dietary pattern using a linear programming algorithm we developed (Fig. 2b, Supplementary Methods) and found that, although none of these dietary patterns includes any constraint on amino acid intake, they still differ greatly with each other in the values of amino acid consumption. Moreover, each dietary pattern allowed for substantial flexibility in the intake of all amino acids (maximal daily intake/ minimal daily intake > 20 for all dietary patterns and amino acids, Fig. 2b), revealing the possibility to modulate amino acid intake under a certain dietary pattern.

To quantify the variability of amino acid composition that is independent of energy and protein intake, we developed a sampling algorithm based on the accelerated convergence hit-and-run method[40] to quantify the amino acid composition of each diet by sampling 50,000 instances of each diet (Supplementary Methods). We first confirmed that the sample size of 50,000 was sufficient to capture the distribution of amino acid abundance in a dietary pattern based on the convergence of the sample mean and standard deviation values (Supplementary Fig. 3a). PCA of the sampled diets (Fig. 2c) and comparison of mean values (Fig. 2d) showed that the ten dietary patterns also have different signatures of amino acid composition. Notably, differences in amino acid composition also exist between dietary patterns similar to each other such as the vegetarian diet and plant-based diet. Indeed, we observed a 30% of difference in methionine abundance between vegetarian diet and plant-based diet (0.019 g methionine/g total AAs in vegetarian diet compared to 0.014 in plant-based diet), suggesting that small changes in the choice of foods result in substantial differences in amino acid intake (Fig. 2d, Supplementary Fig. 3b). We also estimated compositions of carbohydrates and fats in these diets (Supplementary Fig. 3c), and quantified the variability of amino acid composition across human diets using F-statistic values from one-way ANOVA, and compared it with the variability of carbohydrates and fats across dietary patterns (Fig. 2e). Strikingly, we found that the variability of amino acid composition across diets was much higher than that of carbohydrates and fats, with the amino acids lysine, methionine, proline and histidine being the most highly variable across human dietary patterns (F-statistic > 50,000 compared to less than 10,000 for carbohydrates and fats, Fig. 2e, f, Supplementary Fig. 3b, c). Among these amino acids, lysine, histidine and methionine are significantly lower in instances of the plant-based diet, and proline is significantly lower in Paleo diet (Fig. 2f). On the other hand, amino acids with the lowest F-statistic values hence lowest variability across dietary patterns are serine (F-statistic = $6.7 \times 10^3$), tyrosine (F-statistic = $8.4 \times 10^3$), and glycine (F-statistic = $9.4 \times 10^3$). Notably, these results were unaffected by log-transforming the amino acid levels, suggesting that the findings about variability of amino acids in foods and diets were robust to changes in scale (Supplementary Fig. 4a-d). To exclude the possible confounding influence of the correlation between the absolute levels of amino acid intake and variability of amino acids in

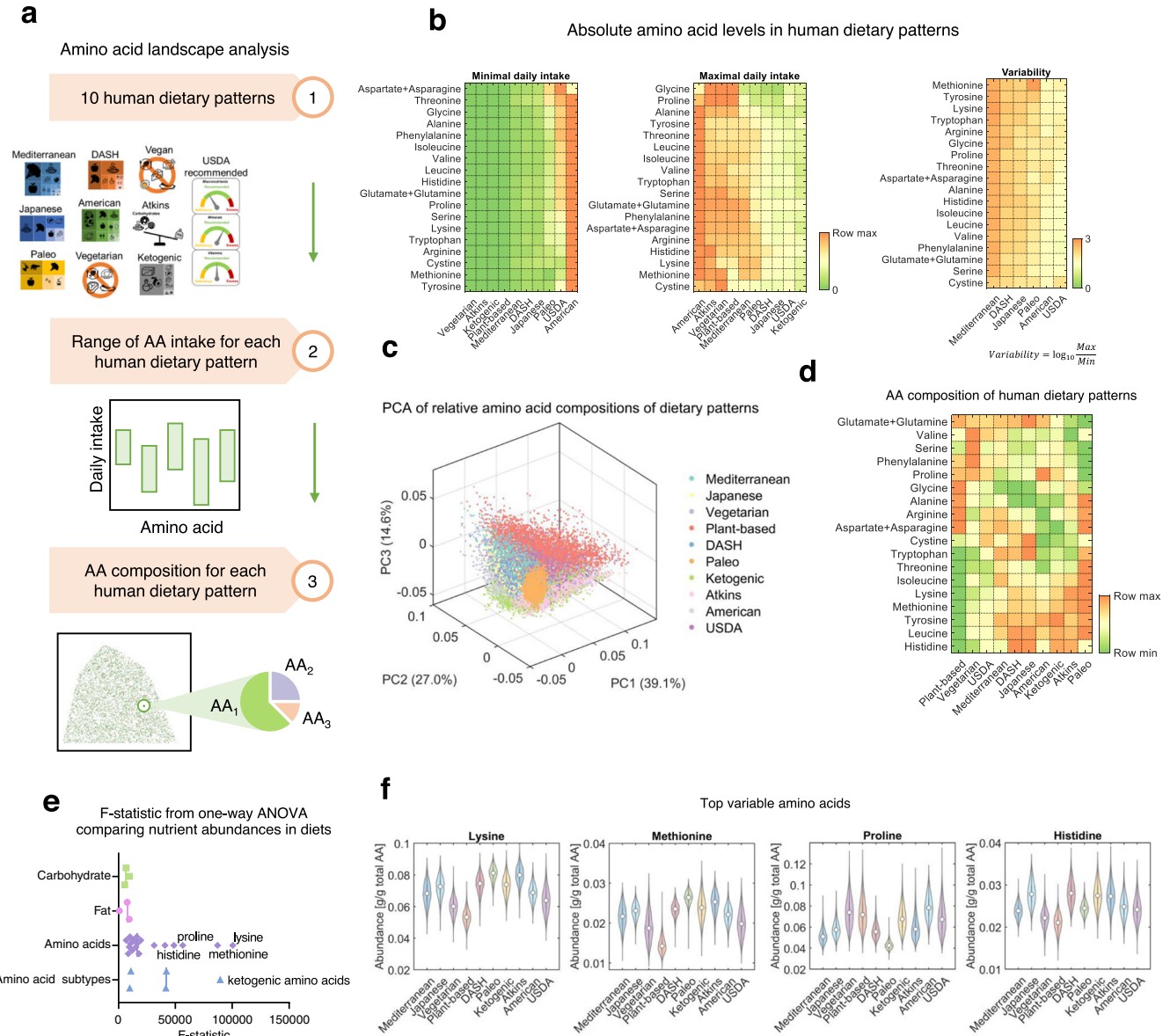

**Fig. 2 | Amino acid landscape of human diets. a** Workflow for the computational modeling of amino acid abundance in human dietary patterns. Food images used with permission from Microsoft. **b** Absolute levels of amino acids in human dietary patterns quantified by the minimal and maximal daily intake values of amino acids in each dietary pattern. **c** Principal components analysis (PCA) of relative amino acid compositions of human diets sampled for all ten dietary patterns. Each dot represents for a diet. Colors of the dots indicate different dietary patterns. **d** Average amino acid composition of the ten human dietary patterns. **e** F-statistic values from one-way analysis of variance (ANOVA) comparing the composition of single amino acids, amino acid subtypes, carbohydrates, and fats across human dietary patterns. **f** Violin plots showing the distributions of amino acids that are the most variable across human dietary patterns. The circles indicate median values.

---

foods and diets, we confirmed that the median level and F-statistic of amino acids were uncorrelated with each other in both human foods and dietary patterns (Pearson's R = −0.20 for foods and 0.02 for dietary patterns, Supplementary Fig. 4e, f). The amino acid signatures of human dietary patterns were further validated by measurements of fasting blood concentrations of the amino acids leucine, isoleucine, and alanine in human subjects eating plant-based or ketogenic diet (Supplementary Fig. 4g, h)[41]. We also confirmed that the amino acid signatures of human dietary patterns were robust to variation in the definitions of dietary patterns by comparing amino acid signatures of Atkins diet and Mediterranean diet computed using different definitions of these diets (Supplementary Fig. 5). Taken together, these results reveal that the biggest difference in macronutrient composition across human dietary patterns is in amino acid content, and not that of carbohydrates or fats. How the diversity in dietary amino acids results in different health outcomes remains an open question, which

may begin to be answered with nutritional and health data in large populations of humans.

### Landscape of amino acid intake in human dietary profiles
Next, we considered individual dietary amino acid intake profiles across a population of individuals from diverse ethnic and cultural backgrounds. We reconstructed the dietary amino acid intake profiles in more than 30,000 human subjects in the United States based on 24-hour dietary recalls in the National Health and Nutrition Examination Survey (NHANES) 2007-2014 datasets (Fig. 3a). Although 24-hour dietary recalls used in NHANES have limitations such as underreporting of energy intake compared to other approaches for dietary assessment such as dietary records, this approach has advantages in large epidemiologic cohorts such as cost-effectiveness[42–44]. Since the NHANES datasets do not direct include dietary amino acid intake values, we developed a set of computational tools for data imputation

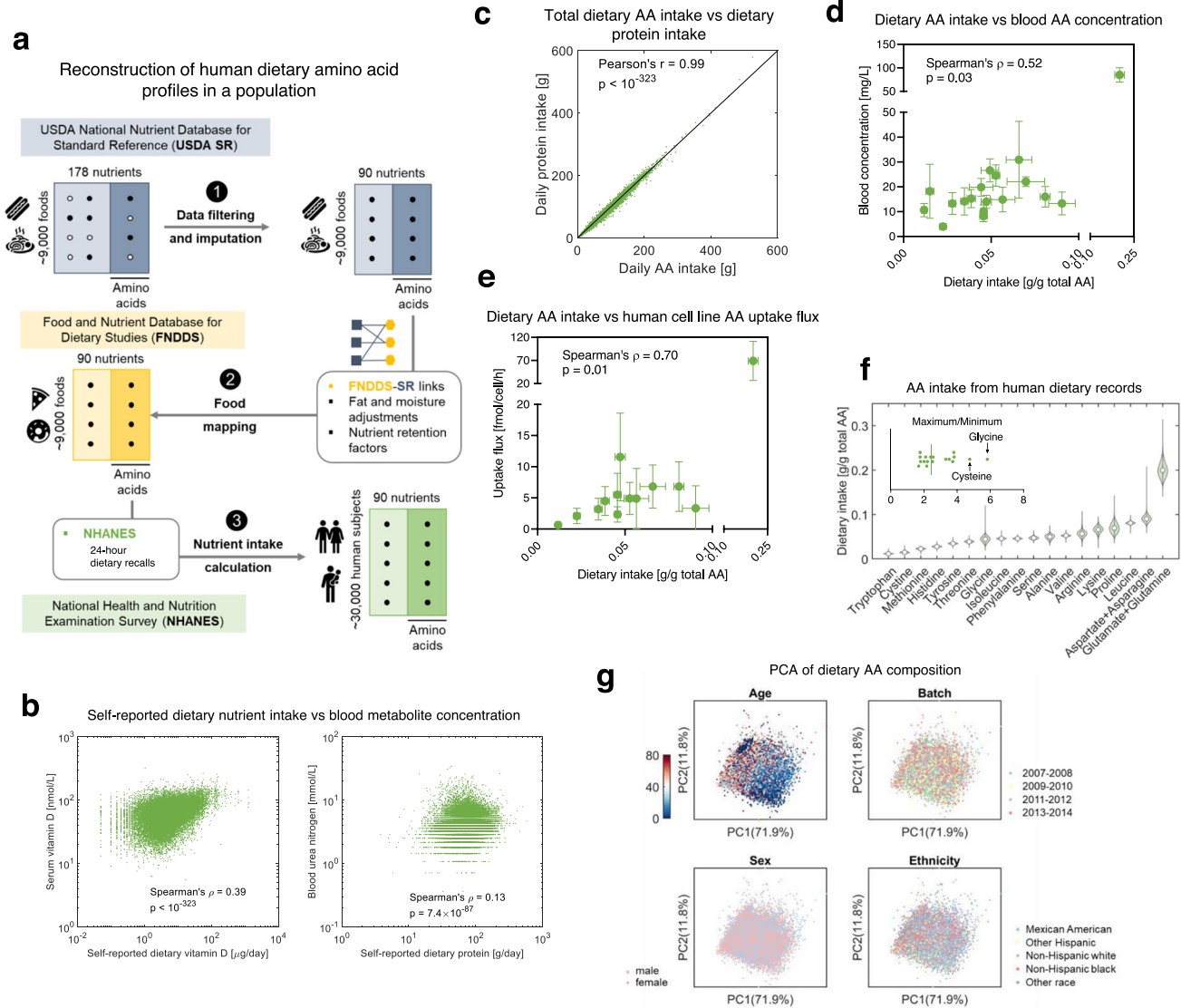

**Fig. 3 | Landscape of human dietary amino acid intake. a** Workflow for reconstruction of the database consisting of amino acid intake profiles in human dietary data. Food images used with permission from Microsoft. **b** Comparison between nutrient intake values in the self-reported dietary data and laboratory measurements of nutrient-related metabolites in blood. A two-sided Spearman's rank correlation test was performed to compute the p-value. **c** Comparison between total dietary amino acid intake in the reconstructed amino acid intake database and dietary protein intake in the original dietary data. A two-sided Pearson's correlation test was used to compute the p-value. **d** Comparison of the reconstructed human dietary amino acid intake values to blood concentrations of amino acids. The dots represent for mean values and error bars for standard deviations. A two-sided

Spearman's rank correlation test was performed to compute the p-value. $n = 30899$ for dietary intake of amino acids, $n = 494$ for blood concentration of amino acids. **e** Comparison of the reconstructed human dietary amino acid intake values to uptake fluxes of amino acids. The dots represent for mean values and error bars for standard deviations. A two-sided Spearman's rank correlation test was performed to compute the p-value. $n = 30899$ for dietary intake of amino acids, $n = 60$ for amino acid uptake fluxes. **f** Distributions of amino acid intake in human dietary intake profiles. The circles indicate median values. **g** Principal components analysis (PCA) of amino acid intake values in human dietary intake profiles showing their association with age, sex, ethnicity, and batch of the data.

and mapping to reconstruct the amino acid profiles for the NHANES dietary data based on two additional datasets, the USDA SR food nutritional database and the Food and Nutrient Database for Dietary Studies (FNDDS) (Fig. 3a). Data imputation using random forest (RF) regression, an algorithm for multiple imputation which outperformed other methods in the accuracy of imputation (Supplementary Fig. 6a, b), was applied to estimate the missing values of amino acid levels in the USDA SR dataset. The imputed datasets were then used to construct amino acid profiles for the FNDDS and NHANES data by mapping foods in the USDA dataset to foods in the FNDDS dataset which were then used to compute nutrient intake values related to the NHANES dietary recalls (Fig. 3a, Supplementary Methods). To assess the limitations of self-reported dietary profiles in the NHANES data[43], we first

compared the distributions of Euclidean distance between matched (i.e. data from the same individual on different days) and unmatched (i.e. data from different individuals) dietary intake profiles to confirm that the dietary intake profiles were consistent between the two 24-hour recalls (Supplementary Fig. 6c, Wilcoxon's rank-sum $p$ value $<10^{-323}$). We then compared our computed nutrient intake values with measurements of blood concentrations of related metabolites such as Vitamin D and found significant positive correlations between self-reported dietary Vitamin D intake and serum concentration of Vitamin D (Spearman correlation = 0.39, $p$ value $<10^{-323}$, Fig. 3b), and between self-reported dietary protein intake and blood concentration of urea nitrogen (Spearman correlation = 0.13, $p$ value $<10^{-323}$, Fig. 3b). Next, to validate the reconstructed amino acid intake levels, we first compared

the total intake of amino acids and intake of protein in each dietary intake profile and confirmed that the reconstructed total amino acid intake closely resembles the known total protein intake (Pearson correlation = 0.99, $p$ value <$10^{-323}$, Fig. 3c). We then correlated the average amino acid intake profile in the NHANES datasets with the concentrations of amino acids in human blood (Spearman correlation = 0.52, $p$-value = 0.03, Fig. 3d), uptake fluxes of amino acids in human cell lines, which reflect demands of amino acids in cultured human cells (Spearman correlation = 0.70, $p$ value = 0.01, Fig. 3e), and amino acid composition of several culture mediums (Spearman correlation > 0.5 and $p$ value <0.05 for 4 out of 7 culture media, Supplementary Fig. 6d). Although measurements of circulating amino acids in the individuals in NHANES were not available and the correlation between dietary intake and circulating levels of amino acids in different individuals could be weak in certain circumstances[45], the high correlation between average dietary amino acid intake and physiological parameters related to amino acids suggests that our reconstructed amino acid intake data may reflect some aspects of physiological metabolism and suggest that the cellular behaviors and tissue microenvironment in amino acid metabolism reflect to some extent dietary intake of amino acids despite the many other factors that influence cellular metabolism. Finally, we compared the amino acid signature of a ketogenic diet predicted using our sampling-based approach and the amino acid intake profiles of individuals in the NHANES dataset that reportedly adhere to a ketogenic diet and found that they were consistent (Spearman correlation = 0.5, $p$ value = 0.03, Supplementary Fig. 6e, Supplementary Methods).

We then evaluated the overall variability in the intake of each amino acid based on the ratio of maximal to minimal intake values in the human dietary profiles (Fig. 3f), and performed PCA on the reconstructed dietary amino acid profiles to report the association between dietary amino acid composition and demographic variables such as age, sex, and ethnicity (Fig. 3g). We found that among the population included in the NHANES 2007-2014 cohorts, daily intake of amino acids typically varies by two to six-fold (e.g. maximal intake/ minimal intake = 4 for tryptophan, 2.5 for methionine, 6.2 for glycine, and so on). Dietary amino acid composition profiles showed no difference between batches (Fig. 3g), thus confirming that our reconstruction is not biased by batch effect. Interestingly, dietary intake of amino acids was found to correlate with age, while no dependency on other demographic variables such as sex and ethnicity was observed (Fig. 3g, Supplementary Fig. 7). These reconstructed dietary amino acid intake profiles allow us to examine the quantitative relationship between dietary amino acids and human health.

## Dietary amino acid intake associations with human health

We next attempted to link dietary amino acid intake and the prevalence of several human diseases based on the reconstructed dietary amino acid intake profiles and clinical data available in the NHANES database. We focused on chronic diseases that are a major concern to human health, such as cardiovascular disease, diabetes, and cancer. We retrieved the medical information of 18,196 adult subjects in the NHANES 2007-2014 datasets and defined quantitative scores describing the prevalence of hypertension, obesity, cancer, and diabetes based on the examination, laboratory, and questionnaire datasets (Fig. 4a, Methods). Although the data of disease prevalence available in NHANES do not distinguish new and pre-existing diseases, they can still provide useful information for identifying the associations between disease burden and dietary intake. We first computed partial Spearman's rank correlation coefficients as a metric to evaluate the association between dietary amino acid composition and the prevalence of each of the four diseases while controlling for confounders, including demographic and lifestyle-related factors (Supplementary Fig. 8). Moreover, to control for total energy and total protein intake, amino acid intake values were normalized to the total intake of all

amino acids. We identified many amino acid intake-disease associations involving all four diseases considered (statistically significant associations in 21 out of 72 amino acid-disease pairs, Fig. 4b, Methods), among which obesity showed the strongest association with dietary amino acid composition (obesity prevalence positively correlated with the intake of histidine, alanine, glycine, lysine and methionine, and negatively correlated with intake of tryptophan, phenylalanine, valine, serine, asparagine, aspartate, glutamine, and glutamate, Fig. 4b). These associations between dietary amino acid intake and obesity were consistent with some observations in molecular studies, such as the anti-obesity functions of dietary tryptophan and pro-obesity functions of methionine in mice[46,47]. As a control, we also correlated the prevalence of the four diseases with dietary intake of different types of carbohydrates and fats. Counterintuitively, we found much fewer statistically significant associations between dietary intake of carbohydrate and fat (9 significant associations out of 40 disease-nutrient pairs, Fig. 4c). These results together highlight the unexpectedly strong association between that dietary intake of amino acids and human disease which exceeds the association for dietary carbohydrates and fats. To further explore these questions, we performed a comparison of the association between nutrients and human health using machine learning models predicting health outcomes from different types of nutritional variables (Fig. 4d). We categorized nutritional variables included in the NHANES database into six groups, including energy, macronutrients, macronutrient compositions (i.e. fractions of different types of carbohydrate and fat in total carbohydrate and fat intake), vitamins, minerals, amino acid compositions (i.e. intake of each amino acid with the unit g/g total AA), and other nutrients. For each disease, the nutritional variables were used together with the potential confounders as covariates to build an elastic net regularized logistic regression model to predict the prevalence of that disease. Survey weights were also considered in this model by integrating the weights in the lost function used in training the model. The area under receiver operating characteristic curve (AUC) with 5-fold cross-validation was used to assess the performance of the models in predicting disease prevalence (Fig. 4e). We then compared the fraction of nutritional variables that affect disease outcomes (i.e. nutritional variables with non-zero regression coefficients in the machine learning models) in amino acid composition, macronutrient composition, and macronutrient levels as indicators of the importance of that group of nutritional variables in affecting disease prevalence (Fig. 4f). We found that the prevalence of all four diseases can be predicted from nutrient intake (AUC > 0.6 for all diseases, Fig. 4e) and dietary amino acid composition affected the prevalence of all four diseases (each of the four diseases was affected by at least 20% of the amino acid variables, Fig. 4f), thereby further supporting the conclusion that the association between dietary amino acid intake and disease outcomes exceeds the association for dietary carbohydrates and fats. Furthermore, we also quantified the importance of each variable in the machine learning model using standardized regression coefficients to assess its contribution in determining the health outcomes. The computed variable importance for amino acids were comparable to or higher than those for dietary carbohydrates and fats (Supplementary Fig. 9a). We also trained the model using log-transformed variables and found that the transformation had little impact on the regression coefficients (Supplementary Fig. 9b) and AUC values of the model (Supplementary Fig. 9c). Taken together, these results suggest that dietary amino acid intake has strong association with the prevalence of several diseases, thereby providing a rationale for optimization of dietary amino acid intake.

## Guidelines for dietary amino acids and diet design

Dietary recommendations, such as these in the USDA Dietary Guidelines for Americans, often involve suggestions to consume a variety of minimally processed foods and recommended ranges

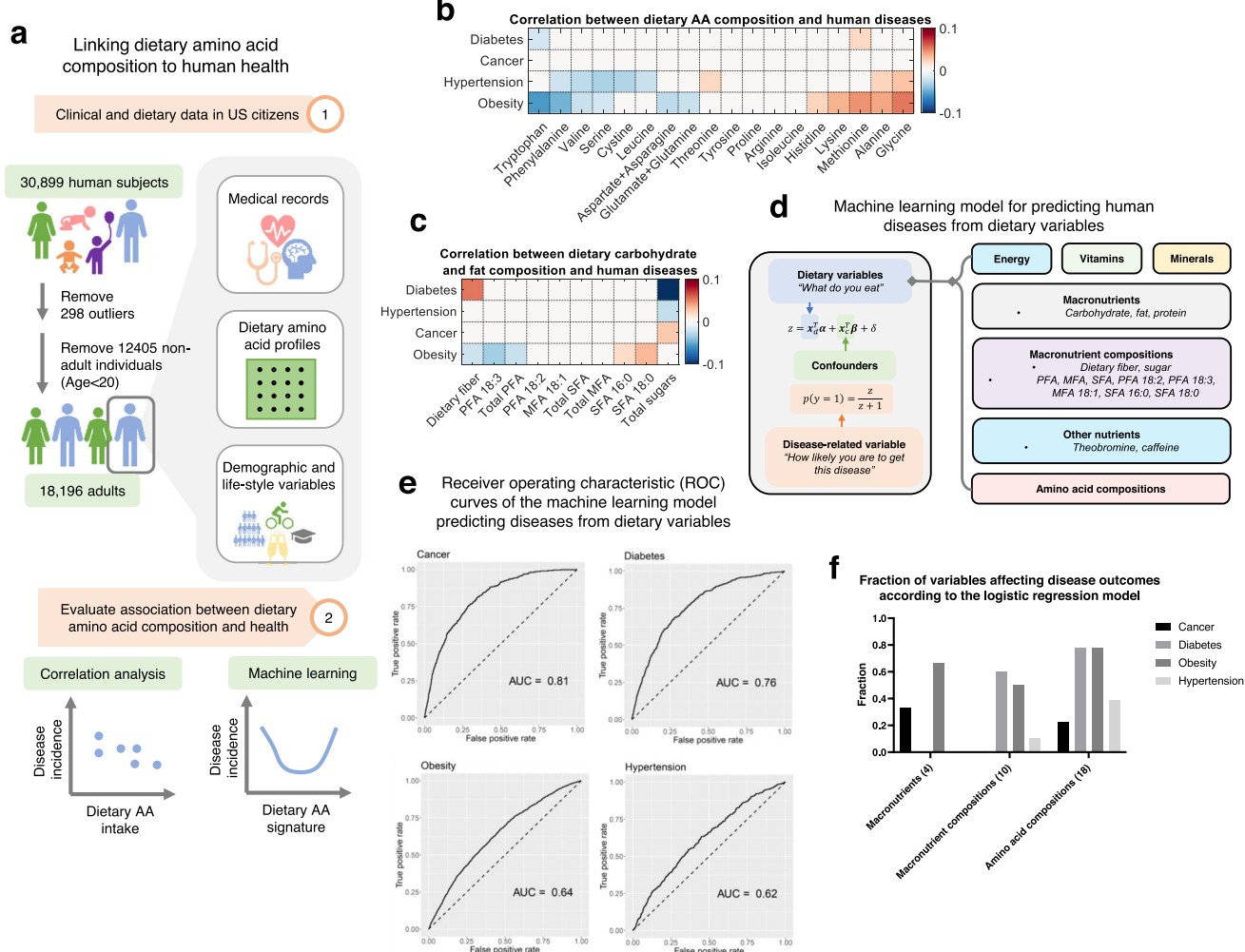

**Fig. 4 | Amino acid intake is predictive of human health. a** Workflow for the analysis of association between dietary amino acid intake and human health. Images used with permission from Microsoft. **b** Partial Spearman correlation between prevalence of human diseases and dietary intake of amino acids. **c** Partial Spearman correlation between prevalence of human diseases and dietary intake of different types of carbohydrate and fat. **d** Framework of the machine learning model predicting prevalence of human diseases from different groups of dietary variables. **e** Receiver operating characteristic (ROC) curves of the machine learning models predicting the four disease outcomes from dietary variables. **f** Fraction of nutritional variables that affect the disease outcomes in macronutrients, macronutrient compositions, and amino acid compositions.

for intake of nutrients, including macronutrients, vitamins, and minerals. Since dietary intake of amino acids has been associated with health outcomes both in molecular studies and by our analysis thus far, we sought to develop an Artificial Intelligence (AI)-based approach for the identification of dietary guidelines for amino acids and design of personalized human diets optimizing their amino acid composition.

First, we developed an algorithm for the identification of amino acid intake guidelines based on the associations between dietary amino acid intake and human health (Fig. 5a). We first focused on obesity since it had the highest prevalence among all four diseases and was found to have the strongest association with dietary amino acid intake among the four diseases considered in this study (Fig. 4b). We identified two categories of obesity-associated amino acids according to the corresponding partial Spearman correlation coefficients and regression coefficients (Fig. 5b), including amino acids for which the intake positively associate with obesity prevalence ('positive association', defined by positive Spearman correlation and positive regression coefficient), and those negatively associate with obesity prevalence ('negative association', defined by negative Spearman correlation and negative regression coefficient). The amino acids phenylalanine, tryptophan, and valine fell into the

negative association group. On the other hand, the amino acids glycine and methionine were categorized into the positive association group. The association between dietary intake of amino acids and obesity was not due to changes in calorie intake, since amino acids positively or negatively associated with obesity were not those with highest correlation coefficients with calorie intake (Supplementary Fig. 10a).

We also examined whether there exists a dietary pattern that can minimize the intake of the amino acids positively associated with obesity while maximizing the intake of the amino acids negatively associated with obesity. To our surprise, no dietary pattern was able to satisfy all of these requirements. For instance, the plant-based diet has the lowest levels of methionine, which positively associates with obesity. Nevertheless, the plant-based diet also has the lowest intake of tryptophan, which negatively associates with obesity, hence cannot satisfy the requirements of maximizing amino acids negatively associated with obesity and minimizing amino acids positively associated with obesity simultaneously. These results reveal the complexity in the relationship between dietary amino acid intake and obesity, indicating trade-offs between the goals of maximizing or minimizing different groups of amino acids which should be considered while designing dietary guidelines for amino acids.

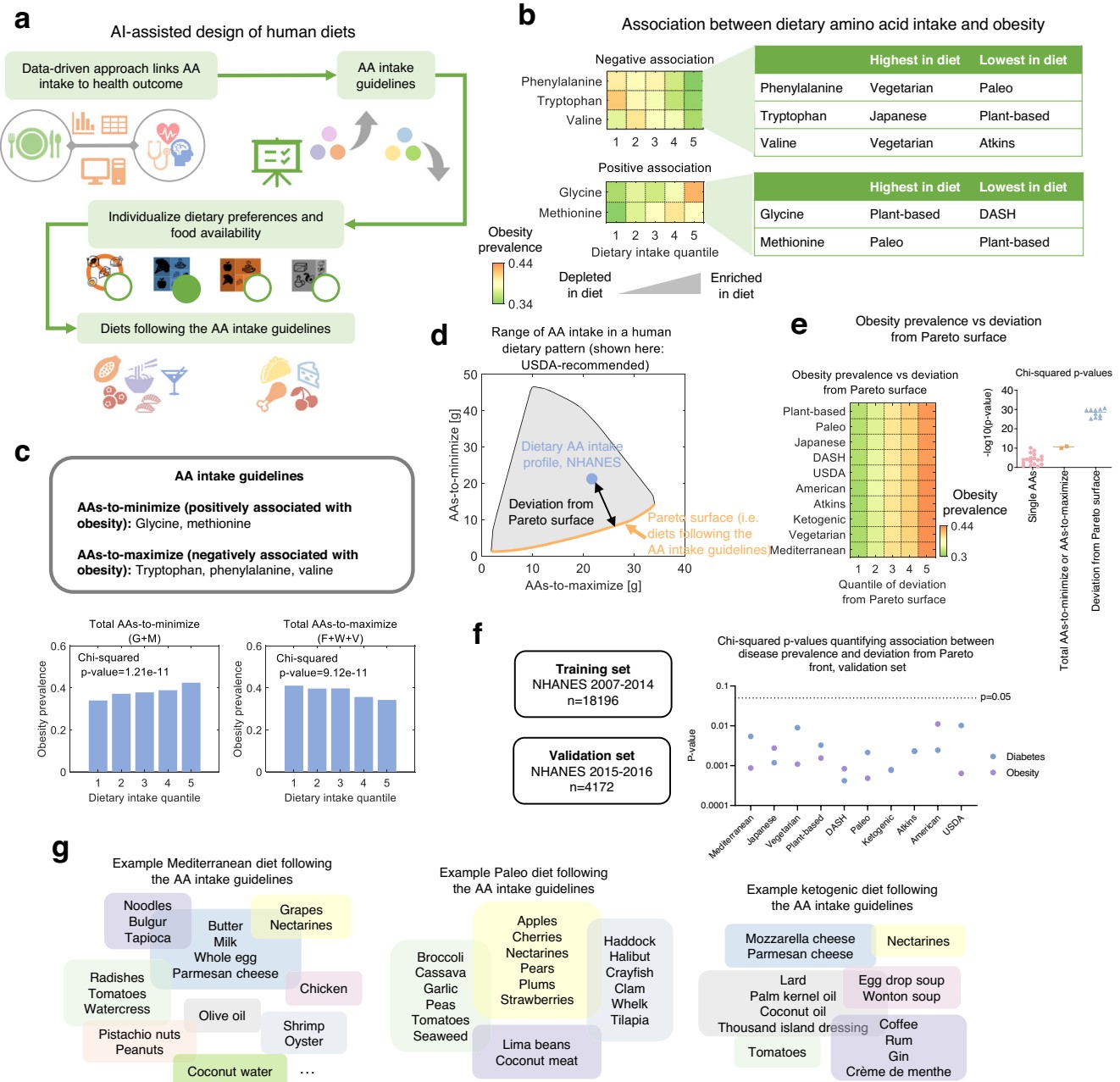

**Fig. 5 | AI for dietary amino acid guidelines and personalized diet design.**
**a** Workflow for AI-assisted identification of dietary amino acid guidelines and design of personalized diets. Images used with permission from Microsoft. **b** Amino acids that positively or negatively associate with obesity prevalence in humans. **c** Identification and confirmation of amino acid intake guidelines based on the association between dietary amino acids and obesity. The p-values were calculated by two-sided chi-squared test. **d** Ranges of intake of total amino-acids-to-maximize and amino-acids-to-minimize in the dietary pattern of the United States of America Department of Agriculture (USDA) -recommended diet (grey shaded region) and the Pareto surface (orange bold curve) corresponding to the two guidelines, i.e.

maximizing total amino-acids-to-maximize, and minimizing total amino-acids-to-minimize. **e** Associations between the obesity prevalence and deviation of dietary intake profiles from the Pareto surface. P-values were calculated by two-sided chi-squared test to assess the significance levels of the associations. $n = 18$ for single amino acids, $n = 2$ for AAs-to-maximize or AAs-to-minimize, $n = 10$ for deviation from Pareto surface. **f** Chi-squared p-values quantifying the association between deviation from Pareto surface and prevalence of obesity and diabetes in the validation set. The p-values were calculated by two-sided chi-squared test. **g** Examples of diets designed according to the amino acid intake guidelines and personalized preferences of dietary patterns.

We therefore sought to define dietary amino acid intake guidelines based on the association between dietary amino acids and obesity (Fig. 5c), that is, to minimize the total intake of amino acids that positively associate with obesity (i.e. AAs-to-minimize, including glycine and methionine), and to maximize the total intake of amino acids that negatively associate with obesity (i.e. AAs-to-maximize, including tryptophan, phenylalanine and valine). We first confirmed that both total AAs-to-minimize and total AAs-to-maximize were significantly

associated with obesity prevalence (Chi-squared $p$ value = $1.21 \times 10^{-11}$ for total AAs-to-minimize and $9.12 \times 10^{-11}$ for total AAs-to-maximize, Fig. 5c).

We then further characterized the trade-off between the requirements of minimizing total AAs-to-minimize and maximizing total AAs-to-maximize by constructing the Pareto surface based on the two requirements (Fig. 5d). The concept of Pareto optimality has been widely applied in economics and engineering, and introduced to

biology to characterize the trade-off between multiple tasks of bacteria, cancer cells, and organisms[48–51]. For each dietary pattern, there exists a Pareto surface consisting of diets that best balance the needs to minimize total AAs-to-minimize and to maximize total AAs-to-maximize, meaning that for a diet within the Pareto surface, any other diet following this dietary pattern would never have both higher total intake of AAs-to-maximize and lower total intake of AAs-to-minimize at the same time. We hence developed an algorithm to construct the Pareto surface for each of the ten dietary patterns considered in this study (Fig. 5d, Supplementary Fig. 10b, Methods), and quantified the extent by which a specific diet satisfies the two requirements of maximizing total AAs-to-maximize and minimizing total AAs-to-minimize using the deviation from Pareto surface (Fig. 5d). For each dietary pattern, we computed the deviation of each NHANES dietary intake profile from its Pareto surface, and found that the deviation from the Pareto surface strongly correlates with obesity prevalence (Chi-squared $p$ values $<10^{-20}$ for all dietary patterns, Fig. 5e), implying that diets on the Pareto surface of each dietary pattern are associated with lower risk of obesity. On average, an individual that eats a diet that is the top 20% furthest away from the Pareto surface has a 38% higher chance of being obese compared to one eating a diet among the 20% closest to the Pareto surface (Fig. 5e).

To test if a similar association between disease prevalence and deviation from the Pareto surface also exists for other diseases, we applied this approach to identify amino acids positively and negatively associated with diabetes (Supplementary Fig. 10c) and constructed the Pareto surfaces based on those diabetes-associated amino acids. We found that deviation from the Pareto surface defined based on those amino acids also correlates with diabetes prevalence (Supplementary Fig. 10d). Finally, we used the NHANES 2015-2016 dataset, which was not used in learning the relationship between dietary amino acids and disease outcomes, as an external validation of the association of obesity and diabetes with the deviation from their corresponding Pareto surfaces (Chi-squared p-value <0.05 for both diabetes and obesity and all 10 diets, Fig. 5f, Supplementary Fig. 10e-g).

These findings not only reveal novel relationship between dietary amino acid intake and health, but also allow us to design diets that have amino acid profiles associated with lower risk of obesity and satisfy personalized needs and requirements such as preferred dietary patterns according to the constructed Pareto surface of the preferred dietary pattern. Hence, based on such strategy, we developed an AI for designing diets including the Mediterranean, Paleo, and ketogenic diet (Fig. 5g). Starting from a user-defined dietary pattern, the AI searches all combinations of foods that satisfy the requirements of that dietary pattern for diets (i.e. combinations of foods) that lie on the Pareto front determined by the two objectives of minimizing the total intake of amino acids positively associated with obesity and maximizing the total intake of amino acids negatively associated with obesity. Each diet contains a variety of foods from diverse sources and keeps the features of the corresponding dietary pattern.

## Discussion

This study develops data resources and computational techniques to begin to address two major limitations in the nutritional sciences: 1) the lack of systematic collections of nutritional information and 2) the lack of computational tools to probe the connections in food, dietary patterns and practices, and health status. Consequentially, we made a number of findings about the variability of amino acids across different types of human foods and dietary patterns and the unexpected associations between dietary amino acid intake, food and dietary patterns, and health. Unexpected links from amino acid intake to pathology such as obesity highlight non-intuitive diet-disease associations and inherent trade-offs in amino acid content in food.

While we were able to use the tools we devised to study and make discoveries about the landscape of amino acid intake, these capabilities are generalizable to any systematic analysis of human food and diet. For instance, it is still unclear how dietary patterns and human dietary records differ with each other in micronutrients such as vitamins, minerals, dietary fiber, added sugars, and how personalized diets can be designed to cover more nutritional goals. The application of the algorithms we developed in this study may help address these questions.

This study has some limitations. First, the association between dietary amino acids and human diseases is observational and does not directly imply causality. Nevertheless, some amino acid-disease associations identified by our analysis have been observed in experimental studies. For instance, tryptophan, which was found to be negatively associated with obesity in our study, was shown in mice to reduce appetite and weight gain through the production of serotonin in brain[46]. On the other hand, dietary restriction of methionine in mice and human has been shown to improve metabolic health and increase fat oxidation, which may contribute to the anti-obesity effects of dietary methionine restriction[47,52,53]. However, given the complexity of nutritional regulation of human health and the collinearity between intake of nutrients, our analyses have limitations in that they are exploratory and unable to obtain causal relationships between dietary amino acids and human health. Other factors that may affect how dietary amino acids impact human health, such as digestion and absorption of protein and amino acids, are also not considered in our analyses because physiological data quantifying these aspects at the population level are still largely missing. Further studies, such as randomized controlled trials that directly compare the health outcomes of diets differing with each other in amino acids, are necessary but also limited to the cohort in consideration and the pre-determined end points.

Second, the computational analyses of amino acid landscape in human dietary patterns were performed using a sampling-based approach that generates random diets following a specific dietary pattern. The rationale for this strategy is that the abundances of amino acids are uniformly distributed in the feasible region defined by the constraints related to that dietary pattern. Moreover, the sampling approach does not explicitly treat the coupling between intake of amino acids in human diets as constraints. The correlation between amino acid intake sampled by our algorithm is a direct consequence of the geometry of the feasible region. Thereby, it is worth noting that the distribution of intake of single amino acids and the correlation between intake of different amino acids predicted by this method could be different from the actual distribution of amino acid intake in people adhering to the corresponding dietary pattern.

We also note that the datasets used in this study are not completely free of bias. The majority of entries in the databases of foods and human dietary data are western, while foods frequently consumed in other geographical regions and by other cultural groups, such as Asians and Africans, are largely underrepresented. Therefore, application of our findings to non-western populations may be limited. Nevertheless, we are optimistic that this limitation could be addressed by extending the coverage of the existing nutritional and epidemiological datasets to non-western populations[54,55].

## Methods

### Acquisition of datasets

Microsoft Access database files for USDA National Nutrient Database for Standard Reference (SR) and the USDA Food and Nutrient Database for Dietary Studies (FNDDS) were downloaded from the website for USDA Agricultural Research Service: https://www.ars.usda.gov/northeast-area/beltsville-md-bhnrc/beltsville-human-nutrition-

research-center/methods-and-application-of-food-composition-laboratory/mafcl-site-pages/sr17-sr28/ (SR), and https://www.ars.usda.gov/northeast-area/beltsville-md-bhnrc/beltsville-human-nutrition-research-center/food-surveys-research-group/docs/fndds-download-databases/ (FNDDS). SAS (.xpt) files for NHANES 2007-2008, 2009-2010, 2011-2012, 2013-2014, 2015-2016 datasets, including demographics data, dietary data, examination data, laboratory data and questionnaire data were retrieved from https://wwwn.cdc.gov/nchs/nhanes/Default.aspx, and converted to R data frames using the function 'sasxport.get()' in the R package 'Hmisc'. More details about procedures for pre-processing of the data are described in Supplementary Methods.

## Computer algorithms and their implementation

Details about the computer algorithms used in this study, including these for reconstruction of amino acid landscape in human foods, dietary patterns, and dietary intake profiles, are explained below. A full description of the methodology is provided in the Supplementary Methods. The regularized logistic regression model and algorithms for imputation and reconstruction of amino acid profiles in the NHANES database, including imputation of missing data, and mapping of foods in the USDA SR, FNDDS, and NHANES databases, were implemented in R. All other algorithms used in this study were implemented in MATLAB. The database for amino acid abundance in human foods, dietary patterns and dietary intake profiles was implemented in both Microsoft Access database file and Microsoft Excel files. All database files are freely available for download at the GitHub repository: https://github.com/ziweidai/AA_human_diet/tree/main/6-Database.

## Mathematical definition of diets

A diet is defined as a combination of foods consumed by an individual on a daily basis, and the consumed amount of each food included in this diet. A subset of foods (2335 foods in total) in the USDA standard reference food composition database release 28 (USDA SR28) was considered in defining diets. Other foods were discarded in this analysis due to missing values in important nutrients such as carbohydrate, fat, protein, vitamins, minerals, and amino acids. Thus, each diet is defined as a numeric vector with 2335 elements, each of which describes the amount of the corresponding food in this diet:

$$\boldsymbol{x} = \begin{bmatrix} x_1 \\ x_2 \\ \vdots \\ x_{2335} \end{bmatrix} \quad (1)$$

## Mathematical definition of human dietary patterns

Human dietary patterns are defined mathematically as a set of constraints on the composition of foods or nutrients in a diet following that dietary pattern. Three types of constraints are considered in defining a human dietary pattern: constraints on consumption of foods, constraints on absolute intake of nutrients, and constraints on ratio between intake of different nutrients. Detailed descriptions of how these constraints are derived are in the Supplementary Methods. Briefly, a general mathematical form for definition of a human dietary pattern is below:

$$\begin{cases} \boldsymbol{x} \geq \boldsymbol{0} \\ \boldsymbol{l}_n \leq \mathbf{C}\boldsymbol{x} \leq \boldsymbol{u}_n \\ \boldsymbol{l}_f \leq \mathbf{D}\boldsymbol{x} \leq \boldsymbol{u}_f \\ \mathbf{E}\boldsymbol{x} \leq \boldsymbol{0} \end{cases} \quad (2)$$

In which $\boldsymbol{x} \geq \boldsymbol{0}$ is the constraint that consumption of each food is nonnegative, $\boldsymbol{l}_n \leq \mathbf{C}\boldsymbol{x} \leq \boldsymbol{u}_n$ is the constraint on consumption of foods,

$\boldsymbol{l}_f \leq \mathbf{D}\boldsymbol{x} \leq \boldsymbol{u}_f$ is the constraint on absolute values of nutrient intake, $\mathbf{E}\boldsymbol{x} \leq \boldsymbol{0}$ is the constraint on ratio between intake of different nutrients.

## Quantification of amino acids in human dietary patterns

Ranges of absolute amino acid levels in human dietary patterns defined by (2) were determined by solving the two linear programming problems below:

$$\min \boldsymbol{a}_i^T \boldsymbol{x}, \text{s.t.}$$

$$\begin{cases} \boldsymbol{x} \geq \boldsymbol{0} \\ \boldsymbol{l}_n \leq \mathbf{C}\boldsymbol{x} \leq \boldsymbol{u}_n \\ \boldsymbol{l}_f \leq \mathbf{D}\boldsymbol{x} \leq \boldsymbol{u}_f \\ \mathbf{E}\boldsymbol{x} \leq \boldsymbol{0} \end{cases} \quad (3)$$

$$\max \boldsymbol{a}_i^T \boldsymbol{x}, \text{s.t.}$$

$$\begin{cases} \boldsymbol{x} \geq \boldsymbol{0} \\ \boldsymbol{l}_n \leq \mathbf{C}\boldsymbol{x} \leq \boldsymbol{u}_n \\ \boldsymbol{l}_f \leq \mathbf{D}\boldsymbol{x} \leq \boldsymbol{u}_f \\ \mathbf{E}\boldsymbol{x} \leq \boldsymbol{0} \end{cases} \quad (4)$$

The solutions of (3) and (4) give the lower and upper bounds of the intake of the $i$-th amino acid in the dietary pattern. Amino acid composition of a dietary pattern (i.e. intake of each single amino acid relative to the total intake of all amino acids) was estimated by uniformly sampling 50,000 random diets under that dietary pattern using a modified hit-and-run sampling algorithm (details described in the Supplementary Methods) to simulate the probability distribution of amino acid composition in that dietary pattern.

## Imputation of missing data

Missing data imputation for the USDA SR and FNDDS datasets was performed using the random forest algorithm implemented in the R package 'missForest'. To adjust for the collinearity between amino acids and protein in human foods, absolute levels of amino acids were transformed by normalizing to one plus the level of protein in each food:

$$\widehat{Y_{\text{AA}}} = \frac{Y_{\text{AA}}}{1 + Y_{\text{protein}}} \quad (5)$$

The transformed amino acid variables were then used together with absolute levels of other nutrients as the input to the missing data imputation algorithm. After imputation has been done for the transformed variables $\widehat{Y_{\text{AA}}}$, imputation for absolute levels of amino acids can be calculated from the imputed $\widehat{Y_{\text{AA}}}$ values:

$$Y_{\text{AA}}^{(i)} = \widehat{Y_{\text{AA}}^{(i)}}(1 + Y_{\text{protein}}^{(i)}) \quad (6)$$

In which the superscript (i) indicates imputed variables. After data imputation with the USDA SR datasets, nutritional composition values of foods in the FNDDS datasets, which were further used to compute dietary intake of nutrients in the NHANES data, were computed using the imputed USDA SR datasets together with the mapping information from foods in SR to foods in FNDDS with factors for moisture and fat adjustments and retention factors for nutrients considered.

## Definition of disease variables

Binary disease variables indicating the presence of pathological conditions, including obesity, hypertension, diabetes, and cancer, were constructed based on the datasets 'Examination data', 'Laboratory data', and 'Questionnaire data' in the NHANES databases

2007-2008, 2009-2010, 2011-2012, 2013-2014, and 2015-2016. Adults with BMI values higher than 30 were considered obese. Hypertension was defined as the condition of systolic blood pressure (the variables 'bpxsy1', 'bpxsy2' and 'bpxsy3', corresponding to three consecutive measurements) being higher than 120 mm Hg and diastolic blood pressure (the variables 'bpxdi1', 'bpxdi2' and 'bpxdi3', corresponding to three consecutive measurements) being higher than 80 mm Hg. Diabetes was defined as the condition of glycohemoglobin levels (the variable 'lbxgh') being higher than 6.5%, fasting plasma glucose concentration (the variable 'lbxglu') higher than 126 mg/dL, and blood glucose concentration in response to oral glucose tolerance test (the variable 'lbxglt') higher than 200 mg/dL. Information about the presence of cancer was obtained from answers to the question 'Have you ever been told by a doctor or other health professional that you had cancer or a malignancy of any kind?' in the questionnaire about medical conditions, in which the answers 'yes' or 'no' were linked to the presence or absence of cancer, while the answers 'refused' and 'don't know' were considered as missing data.

### Identification of amino acids associated with diseases

A logistic regression model with elastic net regularization of the regression coefficients was built to predict disease prevalence from dietary variables. Under the assumption that the interactions between dietary variables and potential confounders are additive, the model has the form below:

$$p(y=1|\boldsymbol{x}_{AA},\boldsymbol{x}_{nut},\boldsymbol{x}_c) = \frac{e^{(\boldsymbol{w}_{AA}^T\boldsymbol{x}_{AA}+\boldsymbol{w}_{nut}^T\boldsymbol{x}_{nut}+\boldsymbol{w}_c^T\boldsymbol{x}_c+b)}}{1+e^{(\boldsymbol{w}_{AA}^T\boldsymbol{x}_{AA}+\boldsymbol{w}_{nut}^T\boldsymbol{x}_{nut}+\boldsymbol{w}_c^T\boldsymbol{x}_c+b)}} \quad (7)$$

This model links dietary amino acid composition ($\boldsymbol{x}_{AA}$), other dietary variables ($\boldsymbol{x}_{nut}$), and potential confounders ($\boldsymbol{x}_c$) to the disease outcome ($y$, value 1 means that the individual has that disease). The R package 'glmnet' was used to train the model and assess its performance using 5-fold cross validation with the survey weight of each sample in the NHANES dataset taken into consideration. Feature importance for each variable in $\boldsymbol{x}_{AA}$, $\boldsymbol{x}_{nut}$ and $\boldsymbol{x}_c$ was computed by absolute value of the standardized regression coefficient (i.e. the product of the original regression coefficient and the standard deviation of the variable). Partial Spearman's rank correlation coefficients were also computed as an additional metric quantifying the association between dietary variables and disease prevalence. Amino acids with positive partial Spearman's rank correlation coefficients and positive regression coefficients with a disease were considered positively associated with that disease, and those with negative partial Spearman's rank correlation coefficients and negative regression coefficients with that disease were considered negatively associated with that disease.

### Analysis of Pareto optimality

After identification of amino acids positively or negatively associated with a disease, the general mathematical form of optimizing the two amino acid intake goals of minimizing total intake of amino acids positively associated with that disease and maximizing total intake of amino acids negatively associated with that disease under a dietary pattern is defined as shown below:

$$\max \boldsymbol{a}_+^T\boldsymbol{x}, \min \boldsymbol{a}_-^T\boldsymbol{x}, \text{s.t.}$$

$$\begin{cases} \boldsymbol{x} \geq 0 \\ \boldsymbol{l}_n \leq \mathbf{C}\boldsymbol{x} \leq \boldsymbol{u}_n \\ \boldsymbol{l}_f \leq \mathbf{D}\boldsymbol{x} \leq \boldsymbol{u}_f \\ \mathbf{E}\boldsymbol{x} \leq 0 \end{cases} \quad (8)$$

A feasible solution $\boldsymbol{x}_0$ of this problem is defined as a Pareto solution if for any other feasible solution $\boldsymbol{x}_1$, $\boldsymbol{a}_-^T\boldsymbol{x}_1 > \boldsymbol{a}_-^T\boldsymbol{x}_0$ if $\boldsymbol{a}_+^T\boldsymbol{x}_1 > \boldsymbol{a}_+^T\boldsymbol{x}_0$, and $\boldsymbol{a}_+^T\boldsymbol{x}_1 < \boldsymbol{a}_+^T\boldsymbol{x}_0$ if $\boldsymbol{a}_-^T\boldsymbol{x}_1 < \boldsymbol{a}_-^T\boldsymbol{x}_0$. The Pareto surface consisting of all solutions with Pareto optimality was then constructed using the $\varepsilon$-Constraint algorithm to calculate 100 different solutions uniformly distributed in the Pareto surface. Deviation of a diet from the Pareto surface was then computed by calculating the shortest distance between the diet and the 100 Pareto solutions in terms of the total daily intake of AAs-to-maximize and AAs-to-minimize.

### Statistical analysis

Principal component analysis was performed using the MATLAB built-in function 'pca()'. One-way ANOVA was performed using the MATLAB built-in function 'anova1()'. Elastic net regularized logistic regression models were constructed, trained, and evaluated using the functions 'cv.glmnet()', 'glmnet()', 'predict()' and 'performance()' in the R packages 'glmnet' and 'ROCR'. Chi-squared test was performed using the MATLAB built-in function 'crosstab()'. Relationships with p-value <0.05 were considered significant. Partial Spearman's rank correlation coefficients were computed using the MATLAB built-in function 'partialcorr()' with p-values adjusted using the Benjamini-Hochberg procedure. Associations with adjusted p-value <0.05 were considered significant. Average amino acid abundances in food categories or dietary patterns were computed using the mean values of amino acid abundances across all foods in that food category or instances in that dietary pattern.

### Reporting summary

Further information on research design is available in the Nature Portfolio Reporting Summary linked to this article.

## Data availability

The dataset containing abundance of nutrients in human foods used in this study is available in the United States of America Department of Agriculture National Nutrient Database for Standard Reference Legacy Release (USDA SR) [https://data.nal.usda.gov/dataset/usda-national-nutrient-database-standard-reference-legacy-release]. Demographic, dietary, examination, laboratory, and questionnaire datasets used in this study are available in the National Health and Nutrition Examination Survey (NHANES) database [https://wwwn.cdc.gov/nchs/nhanes/Default.aspx]. All datasets generated in this study are available at the GitHub page of Ziwei Dai: https://github.com/ziweidai/AA_human_diet. The datasets have also been deposited to Zenodo and can be accessed by the DOI identifier 10.5281/zenodo.7212850 [https://doi.org/10.5281/zenodo.7212850].

## Code availability

All code and scripts generated in this study are available at the GitHub page of Ziwei Dai: https://github.com/ziweidai/AA_human_diet and can be accessed by the DOI identifier https://doi.org/10.5281/zenodo.7212850 [https://doi.org/10.5281/zenodo.7212850].

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

## Acknowledgements

The authors thank all members of the Locasale Lab and Dai Lab, especially Dr. Zhengtao Xiao, Dr. Shiyu Liu, and Yudong Sun, for helpful discussions. We thank the Marc Lustgarten Foundation, the National Institutes of Health (R01CA193256 to JWL), the American Cancer Society (129832-RSG-16-214-01-TBE to JWL), and the National Key Research and Development Program of China (2021YFA0911300 and 2021YFF1201000 to ZD) for their generous support. Support for computational resources from the Duke Compute Cluster and Data Commons Storage is gratefully acknowledged.

## Author contributions

Z.D. and J.W.L. designed the study, wrote and edited the paper. Z.D. developed the algorithms and analyzed the data with input from W.Z.

## Competing interests

J.W.L. advises Restoration Foodworks, Nanocare Technologies and Cornerstone Pharmaceuticals. The remaining authors declare no other competing interests.
