## [Peer Review File · Nature Communications]

Amino acid variability, tradeoffs and optimality in human dietReviewers' comments:

Reviewer #1 (Remarks to the Author):

The authors have mapped the amino acid composition of a large number of foods and several dietary patterns and then related these to individual health risks through data linkage. It is an extremely difficult paper to assess and review in detail, traversing considerable ground and involving numerous numerical and methodological decisions, assertions, and assumptions. There can be no doubting the scale of ambition, however, and as a case study it is laudable and notable – a way forward. However, conclusions are restricted to the population data sets used (notable NHANES), largely ignore the central issue of nutrient interactions, and are without external validation. Below I have broken down my comments, which remain high level, into the major sections of the paper:

1. The amino acid (AA) landscape of ca. 2,000 human foods is described. The authors conclude that a subset of AA vary more between foods than do component fats and carbohydrates, and using PCA, AA patterns can be clustered by different food categories. To some extent this seems self-evident, but it is interesting to have quantified. I wondered to what extent the greater variance apparent across AA simply reflects the greater dimensionality of AA (18) than of fats and carbohydrates analysed?

2. Human dietary patterns are variable in AA content. Ten dietary patterns were analysed and shown to have distinctive AA signatures, and also that there remains room for variability within dietary patterns. Again, this is interesting (although simply reflects different proportions of different foods and food categories). A more interesting question might be to look at which dimensions of AA are relatively invariant across dietary patterns, as this may reflect regulation of intake. I also wondered about the definitions used for the dietary patterns. There is considerable discussion in the literature around what constitutes a Mediterranean diet, for example.

3. Analysis of AA in human dietary records (30,000 records from NHANES) showed considerable variation, which correlated with blood levels. AA intake varied with age but not ethnicity or sex. The latter is interesting. Regarding the former, this is surprising as the relationships between AA intake and circulating levels is complex and not a simple monotonic +ve association in most cases, other than the BCAAs. Again, there is a large literature on this.

4. Individuals' AA intakes associated with health. The authors used linked data for obesity, CVD, diabetes and cancer and related these to estimates of AA intakes derived from NHANES. The main approach was based on single AA correlations, followed by using machine learning. My overarching concern here is that the approach misses the interactions among AA (and indeed other macronutrients). Avoiding interactions is problematic for two reasons: First, foods and dietary patterns involve mixtures, and covariance among components means they cannot be considered one nutrient at a time. Second, many of the physiological and health effects attributed to single nutrients are in fact the result of interactions between nutrients (ratio effects).

5. Having categorised dietary components into 6 categories – energy, macronutrients, vitamins, minerals, AA composition, it was found that AA were predictive of all diseases but cancer - but so too were the other variables, often to similar degrees or better (see Fig 4E). Does the predictive power increase if interactions among and within nutrient categories are included in the models? Other key question which could be addressed: how do AA patterns in plant- vs animal-derived proteins differ? McArthur et al. 2021 (Cell Metabolism) suggest that AA differences are not important, yet the distinction between animal and plant proteins appear frequently in epidemiological associations.

6. Diet design to minimise obesity. The authors took single AA correlations with obesity and split these into 3 categories (monotonic +ve, monotonic -ve, U-shaped). Actually, there are 5 categories – including bell-shaped and no relationship. Why omit these? They then optimised diets using Pareto optimality to balance maximising +ve AA and minimising -ve AA. Not surprisingly, the Pareto front minimised obesity, which left me wondering about circularity. Here external validation is required to ensure the result is not tautological.

7. Finally, the authors developed AI to design diets (food choices) for different dietary patterns. After which I vowed not to adopt a ketogenic diet, given the desirability of crème de menthe.

Reviewer #2 (Remarks to the Author):

The authors conducted a comprehensive analysis of amino acids contained in foods and human diets and their relationships with human diseases. They concluded that human consumption of amino acids is highly dynamic with substantial variability exceeding that of fat and carbohydrates. In addition, while some amino acids were positively associated with disease risk, others were negatively associated with disease risk. They developed computer algorithms to optimize the composition of amino acids in various human diets to improve certain health outcomes.

The concept and methodology of this study are novel since no previous study has systematically evaluated the variability of amino acids in human diets and its implication for human health. However, the main problem is that although many of the analyses are epidemiological in nature, the analytic design and interpretation of the data lack rigor and sound epidemiologic approaches. For example, the authors did not seem to differentiate disease prevalence and incidence in the analysis, and confounding factors, which are the major threat in epidemiologic analyses, were not adequately addressed. Thus, this study would benefit by including epidemiologic expertise. Another problem is that human diets are complex and the selection of foods is more important than the contents of certain nutrients including amino acid composition. A dietician's or nutritionist's perspective would be very helpful for practical nutritional guidance based on evidence from the study.

Detailed comments:

Line 85-87: The authors compared the resulting F-statistic values of amino acids with those of carbohydrates (i.e. dietary fiber and sugar) and fats (i.e. saturated fat, monounsaturated fat, and polyunsaturated fat). This comparison is "unfair" because there are different types of sugars as well as different subtypes of saturated fat and polyunsaturated fats in foods. The variability of subtypes of sugars and fats in different foods is likely to be larger than that of total sugars or total amounts of saturated fat or PUFA.

Line 142-144. "we found that the variability of amino acid composition across diets was much higher than that of carbohydrates and fats, with the amino acids lysine, methionine, proline and histidine¹ being the most highly variable across human dietary patterns." The same comments above also apply here. Another issue is that there are different ways to construct the same dietary patterns, depending on food choices. For example, the traditional Atkins diets contain very high amounts of red meat and other animal products, but a modified Atkins diet included relatively low amounts of red meat, but higher amounts of plant protein foods.

Some Keto diets include very high amounts of animal fats like butter but others include high amounts of coconut oil or even olive oil. It is important to be specific about specific food choices when discussing various human dietary patterns.

Line 157-159. "We reconstructed the dietary amino acid intake profiles in more than 30,000 human subjects in the United States based on dietary records in the National Health and Nutrition Examination Survey (NHANES) 2007-2014 datasets." There might be a misunderstanding about dietary assessment methodology in NHANES. As far as I know, NHANES collected one or two 24-hour dietary recalls rather than dietary records. Dietary records are considered the "gold standard" in dietary assessment in free-living populations but they are extremely burdensome and expensive and thus not typically not feasible. 24-hr recalls are more practical in large surveys like NHANES. The authors need to describe the methodology and its pros and cons more clearly. The authors also need to take advantage of the repeated measures of diets to reduce measurement errors in self-reported amino acid intakes.

Line 201-204. "We retrieved the medical records of 18,196 adult subjects in the NHANES 2007-2014 datasets and defined quantitative scores describing the incidences of hypertension, obesity, cancer, and diabetes based on the examination, laboratory, and questionnaire datasets." It is unclear how the authors did this. Did they actually retrieve the medical records of the participants? Did they actually assess incidences of the diseases? The NHANES dataset contains prevalence rather than incidence of these diseases. Given the nature of the national survey, it is unlikely for researchers to retrieve the medical records of the participants.

Line 204-207. "We first computed partial Spearman's rank correlation coefficients as a metric to evaluate the association between dietary amino acid composition and the incidences of the four diseases while controlling for confounders including demographic and lifestyle-related factors." Again, it is unclear whether the authors looked at disease incidence or prevalence. Spearman's rank correlations are not an appropriate methodology to evaluate the associations between amino acids and disease risk because of the complex relationship between diet and disease. Typically, multivariate logistic or Cox proportional hazards models are used to estimate disease associations in epidemiologic studies. It is also important to control for potential confounding factors using the multivariate models. Besides dietary and lifestyle confounders, it is also important to control for health care access and neighborhood SES.

Line 230-238. "AUC = 0.55 for amino acids compared to 0.55 for macronutrient composition in predicting obesity, and AUC = 0.53 for amino acids compared to 0.52 for macronutrient composition in predicting hypertension." These AUCs are all fairly low and will not be useful in predicting disease outcomes in clinical or public health settings.

Line 306-309. "Hence, based on such strategy, we developed an AI for designing diets including the Mediterranean, Paleo, and ketogenic diet (Figure 5f). Each diet contains a variety of foods from diverse sources and keeps the features of the corresponding dietary pattern." These food choices seem arbitrary and restrictive for individuals who want to follow certain dietary patterns. There should be more flexibility for food choices, allowing for more diversity in not just amino acids but also other macronutrient compositions across various dietary patterns. Obesity is not the best health outcome for optimizing food patterns because of the problem of reverse causation (i.e. people may change their diet as a result of their weight).

Reviewer #3 (Remarks to the Author):

This is an interesting paper showing the variation of amino acids is greater than fat and carbohydrate categories among food and dietary patterns. Also, it studies the association between amino acid intake derived from self-reported data and health outcomes. I have a few comments regarding the model used and the conclusion made in the paper which is summarized by following parts:

(1) Amino acid landscape of human food

-It is unclear whether the raw abundance data or standardized data are used for the PCA analysis. Given the high %variation explained in PC1, it will be of interest to see the loading factor so the PC can be interpreted.

-The unit is amino acid within each gram of food, but it might be interesting to see amino acid within each cal of food given that daily intake will be more related to total energy rather than the total weight of the food.

-plot 1b suggests most amino acid abundance are right-skewed, so it seems the use of ANOVA and PCA shall be after log transformation of the abundance. Or perhaps the relative abundance is less skewed.

(2) Amino acid landscape of human dietary patterns

-Each human dietary pattern is defined by a linear constraint on food intake. However, this might be just a necessary condition for a pattern rather than a sufficient condition. So the range derived from

linear programming (no constraint found) can be too wider than it should be in the real population.

-Based on the definition, there could be overlap between different dietary patterns, getting PCA by sampling uniformly from the possible range lacks good interpretation to my opinion as the real population with the certain dietary patterns are less likely to be uniformly distributed and when some value can be within multiple dietary patterns, the correlation shall be considered.

(3) Amino acid landscape of human dietary records

-Four imputation scenarios are considered and the one with the best 5-fold CV correlation is selected. But it is unclear within each scenario, whether single or multiple imputations are used? It is well known single imputation will not lead to correct statistical inference. More details are needed if multiple imputations are used.

-It is unclear whether and how survey weights are taken into consideration when using NHANES data.

(4) Association of Amino acid and human health

-The self-reported dietary data are known to have systematic bias. The author proposes to use serum Vitamin D and blood urea nitrogen to determine the quality. Given that the food contains vitamin D and amino-acid could be very different, to me only the protein plot in figure 3b is useful and the correlation is very low.

-since it was known the total energy and protein are associated with many diseases, the author shall consider relative abundance as exposure or include total energy or total protein intake as an adjustment

-the AUC shows no difference in the prediction power between amino acid and carbohydrate and fat intake. $AUC < 0.6$ is generally considered poor, so even statistically significant, $AUC = 0.53$ and $AUC = 0.52$ are not really clinical meaningful differences.

-use logistic regression without transformation on skewed dietary variables might face the model-misspecification problems. The goodness of fit tests is recommended.

-not clear whether survey weight are taken into account

Reviewers' comments:

Reviewer #1 (Remarks to the Author):

The authors have mapped the amino acid composition of a large number of foods and several dietary patterns and then related these to individual health risks through data linkage. It is an extremely difficult paper to assess and review in detail, traversing considerable ground and involving numerous numerical and methodological decisions, assertions, and assumptions. There can be no doubting the scale of ambition, however, and as a case study it is laudable and notable – a way forward. However, conclusions are restricted to the population data sets used (notable NHANES), largely ignore the central issue of nutrient interactions, and are without external validation. Below I have broken down my comments, which remain high level, into the major sections of the paper:

We thank the reviewer for the positive remarks and constructive suggestions. To address the concerns raised by the reviewer, we have improved the machine learning model to consider the interactions between nutrients and introduced additional datasets as external validation of our conclusions. We have also revised the text to improve the clarity of the manuscript.

1. The amino acid (AA) landscape of ca. 2,000 human foods is described. The authors conclude that a subset of AA vary more between foods than do component fats and carbohydrates, and using PCA, AA patterns can be clustered by different food categories. To some extent this seems self-evident, but it is interesting to have quantified. I wondered to what extent the greater variance apparent across AA simply reflects the greater dimensionality of AA (18) than of fats and carbohydrates analysed?

We thank the reviewer for raising this concern and completely agree with the reviewer that the dimensionality is an important factor to consider while comparing the variability of two groups of variables.

First, as the reviewer notes, although the abundances of amino acids have been quantified for many human foods, the variability of amino acid composition across different categories of foods has not been comprehensively quantified. Some previous studies also claim that the difference in amino acid composition between different types of foods, e.g. plant-based versus animal-based foods, is not significant (e.g. MacArthur et al, Cell Metab 2021, PMID: 34270927). Hence, we believe that it is important to quantitatively characterize the variability of amino acid pattern in foods, by both quantifying the variability using statistical analyses such as PCA and ANOVA, and comparing the variability of amino acids to that of carbohydrates and fats.

To further support our conclusion on the variability of amino acids in human foods while controlling for the effects caused by the dimensionality, we have collapsed the 18 amino acids into a few categories of amino acids such as essential amino acids (EAAs), nonessential amino acids (NEAAs), branched-chain amino acids (BCAAs) to reduce the dimensionality of amino acid profiles, and performed one-way ANOVA with the collapsed amino acid variables. Such additional analyses offer a 'fairer' comparison of variability across human foods between

amino acids, carbohydrates, and fats. We found that the collapsed amino acid variables also showed variability comparable to or higher than the variability of carbohydrates and fats across human foods and diets, as supported by the F-statistic values from one-way ANOVA (Response Figure 1). We have included the results in the revised manuscript.

2. Human dietary patterns are variable in AA content. Ten dietary patterns were analysed and shown to have distinctive AA signatures, and also that there remains room for variability within dietary patterns. Again, this is interesting (although simply reflects different proportions of different foods and food categories). A more interesting question might be to look at which dimensions of AA are relatively invariant across dietary patterns, as this may reflect regulation of intake. I also wondered about the definitions used for the dietary patterns. There is considerable discussion in the literature around what constitutes a Mediterranean diet, for example.

We thank the reviewer for the insightful comments.

Regarding the comment on amino acids invariant across dietary patterns, these amino acids can be identified based on the F-statistic from one-way ANOVA. Amino acids with the lowest F-statistic values hence lowest variability across dietary patterns are serine ($F\text{-statistic} = 6.7 \times 10^3$), tyrosine ($F\text{-statistic} = 8.4 \times 10^3$), and glycine ($F\text{-statistic} = 9.4 \times 10^3$). We have included this information and discussion on its potential implications in the revised manuscript.

Regarding the comment on definitions of dietary patterns, in the original manuscript the dietary patterns were defined in a literature-based fashion. For each dietary pattern, we refer to the most authoritative publications we could find on that dietary pattern to derive its definition. For instance, the definition of Mediterranean diet was based on the Mediterranean diet pyramid by the Mediterranean Diet Foundation Expert Group (Bach-Faig et al, Public Health Nutr 2011, PMID: 22166184). Given the flexibility in the definition of a dietary pattern as the reviewer noted, it is important to form a representative definition of dietary patterns for us to proceed with the downstream analyses of dietary patterns.

Nevertheless, we agree with the reviewer that the flexibility in the definition of dietary patterns is important to consider. We have therefore performed an additional analysis of robustness in which we computed amino acid profiles in two additional schemes of Mediterranean diet and two additional schemes of Atkins diet to figure out how the definition of dietary patterns affect the resulting amino acid profiles. Definitions of the two additional Mediterranean diet schemes were developed based on: (1) the nine components of the Mediterranean diet score (MDS) which is a quantitative metric widely used in nutritional epidemiology to assess adherence to the Mediterranean diet (Trichopoulou et al, N Engl J Med 2003, PMID: 12826634); (2) the Mediterranean diet pyramid in an earlier version of the dietary guidelines for adults in Greece (Archives of Hellenic Medicine 1999, 16(5):516-524). Definition of the two additional Atkins diet schemes were developed based on: (1) the “ongoing weight loss” phase of the Atkins diet, in which the upper limit of daily carbohydrate intake is 50 grams instead of 20 grams; (2) the Atkins diet with limited consumption of red meat, in which the consumption of red meat was strictly constrained to zero. We performed one-way ANOVA to compare the amino acid profiles across alternative schemes of the same dietary pattern and compared the resulting F-statistic values to those of the comparison across the original 10 dietary patterns and across three dietary patterns randomly selected from the original 10 dietary patterns (Response Figure 2). We found that the variability of amino acid composition was indeed lower across the different schemes of Mediterranean diet and Atkins diet compared to the variability across the original 10 diets, hence confirming that the amino acid signatures of human dietary patterns are robust to changes in the definitions of dietary patterns. We have included these results in the revised manuscript.

3. Analysis of AA in human dietary records (30,000 records from NHANES) showed considerable variation, which correlated with blood levels. AA intake varied with age but not ethnicity or sex. The latter is interesting. Regarding the former, this is surprising as the relationships between AA intake and circulating levels is complex and not a simple monotonic +ve association in most cases, other than the BCAAs. Again, there is a large literature on this.

We thank the reviewer for the insightful comment. Because measurements of circulating AA levels in the individuals in NHANES were not available, it was unpractical for us to directly correlate circulating level of single amino acids with their corresponding intakes in the NHANES cohort. Instead, we correlated the average intake of each amino acid over the entire NHANES cohort to the average circulating levels of amino acids and average amino acid

intake fluxes across human cell lines. We apologize for the confusion and will clarify this in the revised manuscript. We have also included discussion on the relationship between intake and circulating levels for specific amino acids.

4. Individuals' AA intakes associated with health. The authors used linked data for obesity, CVD, diabetes and cancer and related these to estimates of AA intakes derived from NHANES. The main approach was based on single AA correlations, followed by using machine learning. My overarching concern here is that the approach misses the interactions among AA (and indeed other macronutrients). Avoiding interactions is problematic for two reasons: First, foods and dietary patterns involve mixtures, and covariance among components means they cannot be considered one nutrient at a time. Second, many of the physiological and health effects attributed to single nutrients are in fact the result of interactions between nutrients (ratio effects).

We greatly appreciate the helpful suggestion on considering the interactions between amino acids while assessing their relationship with health.

In the original manuscript, the interactions between amino acids were implicitly considered by normalizing each amino acid intake to the total intake of all amino acids to adjust for the strong correlation between amino acids that can be explained by the variation in protein intake. As the reviewer correctly notes, this approach was unable to account for the interactions between amino acids independent of protein intake, which might confound the associations between amino acids and health outcomes. We agree that such interactions should be carefully adjusted using unbiased statistical approaches. This could be done by either building a generalized linear model that takes all nutrients and potential confounders as independent variables and using the linear coefficients to assess the association between nutrients and health (in other words, interactions between the covariates were assumed to be additive) or fitting a model with pairwise interaction terms to precisely quantify the interaction between each pair of variables (i.e. interactions were assumed to be multiplicative) (Greenland, Stat Med 1983, PMID:6359318).

However, compared to the sample size of the NHANES dataset (i.e., the number of human subjects), the dimensionality of the dataset (i.e., the number of nutrients) is too high to allow efficient fitting of a full model containing all possible multiplicative interactions while avoiding overfitting. Hence, in the revised manuscript, we assumed that the interactions between nutrients are additive and thereby built a logistic regression model with elastic net regularization that links dietary amino acids, other dietary variables, and potential confounding factors to health outcomes:

$$p(y = 1 | \mathbf{x}_{AA}, \mathbf{x}_{nut}, \mathbf{x}_c) = \frac{e^{(\mathbf{w}_{AA}^T \mathbf{x}_{AA} + \mathbf{w}_{nut}^T \mathbf{x}_{nut} + \mathbf{w}_c^T \mathbf{x}_c + b)}}{1 + e^{(\mathbf{w}_{AA}^T \mathbf{x}_{AA} + \mathbf{w}_{nut}^T \mathbf{x}_{nut} + \mathbf{w}_c^T \mathbf{x}_c + b)}}$$

In addition to evaluating the predictive power of the model using the area under the receiver operating characteristic (ROC) curve, we also computed feature importance of each variable using standardized regression coefficients to assess its contribution in determining the health

outcomes (Response Figure 3). This approach, while offering an unbiased approach to evaluate the effects of dietary intakes on human health while controlling for confounding and interactions between nutrients, further supports our conclusion that dietary intake of amino acids are important determinants of human health outcomes. We found that the prevalence of all four diseases can be predicted from nutritional intakes ($AUC > 0.6$ for all four diseases, Response Figure 3A) and were all affected by dietary amino acid intake (more than 20% of amino acid variables have non-zero regression coefficients for all four diseases, Response Figure 3B). The computed variable importance for amino acids were also comparable to or higher than those for dietary carbohydrate and fat intake (Response Figure 3B and 3C). We have included these results in the revised manuscript.

5. Having categorised dietary components into 6 categories – energy, macronutrients, vitamins, minerals, AA composition, it was found that AA were predictive of all diseases but cancer - but so too were the other variables, often to similar degrees or better (see Fig 4E). Does the predictive power increase if interactions among and within nutrient categories are included in the models? Other key question which could be addressed: how do AA patterns in plant- vs animal-derived proteins differ? McArthur et al. 2021 (Cell Metabolism) suggest that AA differences are not important, yet the distinction between animal and plant proteins appear frequently in epidemiological associations.

We thank the reviewer for the constructive comment and have confirmed that including interactions between nutrients could improve the predictive power using a regularized logistic

regression model to account for the additive interactions between nutrients (Response Figure 3A).

We thank the reviewer for the comment mentioning the McArthur et al Cell Metabolism 2021 paper because it offers us the opportunity to share our opinion on that paper. Briefly, that paper compared the amino acid profiles between animal- and plant-proteins and found that the difference in mean amino acid levels between the two categories of protein was subtle. This approach neglects the potential variability in amino acid levels between different categories of plant- and animal-derived proteins, thereby oversimplifying the landscape of amino acid profiles in human foods. To further support our opinion, we have compared the amino acid profiles between plant- and animal-based foods (Response Figure 4). We computed the coefficient of variation (CV) as a metric quantifying the variability of amino acid profiles among animal-based, plant-based, and all foods and found that the variability of amino acid abundances among animal-derived and plant-derived foods was comparable to or even higher than that among all foods for all amino acids except for histidine (higher CV among plant-derived or animal-derived protein compared to CV among all foods, Response Figure 4B). We have included these additional results and related discussion in the revised manuscript.

Response Figure 4. Variability of amino acid profiles in plant-based and animal-based foods. (A) Violin plots comparing the distributions of amino acid profiles between plant-based and animal-based foods. **(B)** Comparison of coefficient of variation (CV) for amino acid profiles among plant-based, animal-based, and all foods.

6. Diet design to minimise obesity. The authors took single AA correlations with obesity and split these into 3 categories (monotonic +ve, monotonic -ve, U-shaped). Actually, there are 5 categories – including bell-shaped and no relationship. Why omit these? They then optimised diets using Pareto optimality to balance maximising +ve AA and minimising -ve AA. Not surprisingly, the Pareto front minimised obesity, which left me wondering about circularity. Here external validation is required to ensure the result is not tautological.

We thank the reviewer for raising this concern.

Response Figure 5. Relationship between dietary AA intake, obesity, and diabetes. (A) AAs negatively or positively associated with obesity prevalence. (B) Association of obesity prevalence with total intake of AAs-to-minimize (positively associated with obesity) and AAs-to-maximize (negatively associated with obesity). (C) Association between obesity prevalence and deviation from Pareto surface. (D) Chi-squared p-values quantifying the associations shown in (C). (E) AAs negatively or positively associated with diabetes prevalence. (F) Association between diabetes prevalence and deviation from Pareto surface. (G) Chi-squared p-values quantifying the associations shown in (F).

The reason for ignoring the bell-shaped and no-relationship amino acids in the diet design stems from technical considerations: for amino acids positively or negatively associated with obesity, they could be easily optimized by minimizing or maximizing their intake, resulting in a linear objective function that fits perfectly into the scheme of linear programming – a mathematical problem that can be formalized and solved easily. However, for the amino acids with U-shaped association with obesity, individuals with the lowest probability to get obesity were the ones with the highest or lowest intake of that amino acid. It is thereby difficult to model such relationship with a linear objective function. Nevertheless, we agree with the reviewer that introducing the U-shaped associations here could be confusing. Hence, in the revised manuscript, we have improved the approach used to identify amino acids positively or negatively associated with disease outcomes: if an amino acid has a positive partial Spearman correlation coefficient with the disease and a positive linear coefficient in the regularized logistic regression model predicting the prevalence of that disease (Response Figure 3), that amino acid was then identified as positively associated with the disease. Similarly, an amino acid was identified as negatively associated with a disease if its partial Spearman correlation coefficient with the disease and linear coefficient in the corresponding logistic regression model

were both negative. According to this criterion, the amino acids negatively associated with obesity were phenylalanine, tryptophan, and valine, while amino acids positively associated with obesity were glycine and methionine. Linear combinations of these amino acids were able to predict the prevalence of obesity with higher accuracy than those identified in the previous version of manuscript (Chi-squared p -value = $1.2e-11$ compared to $9.0e-8$ for AAs positively associated with obesity, and $9.1e-11$ compared to $4.9e-10$ for AAs negatively associated with obesity, Response Figure 5B). We have then applied this approach to investigate the association between dietary amino acid intake and diabetes, and found that the prevalence of diabetes was significantly associated with the deviation from Pareto front defined by the two objectives of minimizing total AAs positively associated with diabetes and maximizing total AAs negatively associated with diabetes (Chi-squared p -value < 0.05 for all 10 diets, Response Figure 5).

Response Figure 6.

Associations between disease outcomes and deviation from Pareto surface in an external validation dataset. (A)

Correlation between the actual protein intake and total AA intake in the external validation set. (B) Association between obesity prevalence and deviation from Pareto surface in the external validation set. (C) Same as in (B) but for diabetes. (D) Chi-squared p -values quantifying the associations shown in (B) and (C).

To further validate the relationship between dietary amino acid and diseases, we have also performed additional analysis with an external NHANES dataset, the NHANES 2015-2016 dataset, which was not included in the original analysis. We reconstructed dietary amino acid intake for all dietary records included in the NHANES 2015-2016 datasets, and validated the reconstructed dietary AA intake by correlating total daily AA intake to the known values of total daily protein intake (Response Figure 6A). We then correlated the prevalence of diabetes and obesity to the deviation from Pareto fronts in the external dataset, and found that for both diabetes and obesity, the disease prevalence was still significantly positively associated with deviation from the Pareto fronts defined based on AAs positively or negatively associated with the diseases identified based on the original NHANES 2007-2014 datasets (Chi-squared p -value < 0.05 for both diabetes and obesity and all 10 diets, Response Figure 6B-D). The external validation further support our conclusion that health outcomes can be predicted by the deviation from Pareto fronts determined by the two objectives of maximizing and minimizing

total intake of certain amino acids. We have included all these additional analyses in the revised manuscript.

7. Finally, the authors developed AI to design diets (food choices) for different dietary patterns. After which I vowed not to adopt a ketogenic diet, given the desirability of crème de menthe.

We thank the reviewer for the comment. Personally, we completely agree with the reviewer that a ketogenic diet is not very appealing, given the desirability of crème de menthe, cheesecake and bubble tea.

Reviewer #2 (Remarks to the Author):

The authors conducted a comprehensive analysis of amino acids contained in foods and human diets and their relationships with human diseases. They concluded that human consumption of amino acids is highly dynamic with substantial variability exceeding that of fat and carbohydrates. In addition, while some amino acids were positively associated with disease risk, others were negatively associated with disease risk. They developed computer algorithms to optimize the composition of amino acids in various human diets to improve certain health outcomes.

The concept and methodology of this study are novel since no previous study has systematically evaluated the variability of amino acids in human diets and its implication for human health. However, the main problem is that although many of the analyses are epidemiological in nature, the analytic design and interpretation of the data lack rigor and sound epidemiologic approaches. For example, the authors did not seem to differentiate disease prevalence and incidence in the analysis, and confounding factors, which are the major threat in epidemiologic analyses, were not adequately addressed. Thus, this study would benefit by including epidemiologic expertise. Another problem is that human diets are complex and the selection of foods is more important than the contents of certain nutrients including amino acid composition. A dietician's or nutritionist's perspective would be very helpful for practical nutritional guidance based on evidence from the study.

We thank the reviewer for the positive remarks on the novelty of our work, and the constructive suggestions on the epidemiologic approaches. In the revised manuscript, we have repeated the epidemiologic analyses with improved statistical models that incorporate the survey weights.

Detailed comments:

Line 85-87: The authors compared the resulting F-statistic values of amino acids with those of carbohydrates (i.e. dietary fiber and sugar) and fats (i.e. saturated fat, monounsaturated fat, and polyunsaturated fat). This comparison is "unfair" because there are different types of sugars as well as different subtypes of saturated fat and polyunsaturated fats in foods. The variability of subtypes of sugars and fats in different foods is likely to be larger than that of total sugars or total amounts of saturated fat or PUFA.

We thank the reviewer for raising this concern.

We completely agree that a direct comparison of variability in specific types of saturated fat or PUFA to that of amino acids will be helpful. However, the USDA SR database lacks detailed information of sugars and fats sufficiently informative to support a detailed analysis of many subtypes of sugars and fats. For instance, USDA SR has the information about levels of lactose, sucrose, galactose, fructose, maltose, and glucose in certain foods, but this information is only available for about 25% of the foods that we included in the analysis of variability, which is not sufficient to support an unbiased comparison of variability between these sugars and amino acids.

In the revised manuscript, we have collapsed the 18 amino acids into a few categories of amino acids such as essential amino acids (EAAs), nonessential amino acids (NEAAs), branched-chain amino acids (BCAAs) to reduce the dimensionality of amino acid profiles, and performed one-way ANOVA with the collapsed amino acid variables. Such additional analyses offer a 'fairer' comparison of variability across human foods between amino acids, carbohydrates, and fats. We found that the collapsed amino acid variables also showed variability comparable to or higher than the variability of carbohydrates and fats across human foods and diets, as supported by the F-statistic values from one-way ANOVA (Response Figure 7). We have included the results in the revised manuscript.

Line 142-144. "we found that the variability of amino acid composition across diets was much higher than that of carbohydrates and fats, with the amino acids lysine, methionine, proline and histidine¹ being the most highly variable across human dietary patterns." The same comments above also apply here. Another issue is that there are different ways to construct the same dietary patterns, depending on food choices. For example, the traditional Atkins diets contain very high amounts of red meat and other animal products, but a modified Atkins diet included relatively low amounts of red meat, but higher amounts of plant protein foods.

Some Keto diets include very high amounts of animal fats like butter but others include high

amounts of coconut oil or even olive oil. It is important to be specific about specific food choices when discussing various human dietary patterns.

We thank the reviewer for this comment. In the revised manuscript, we have performed additional analysis for a 'fairer' comparison of the variability in amino acids, carbohydrates and fats across human dietary patterns similar to what we plan to do for the comparison of variability across human foods.

Regarding the comment on flexibility in the definition of dietary patterns, the dietary patterns were originally defined by referring to the most authoritative publications we could find on that dietary pattern. For instance, the definition of Mediterranean diet was based on the Mediterranean diet pyramid by the Mediterranean Diet Foundation Expert Group (Bach-Faig et al, Public Health Nutr 2011, PMID: 22166184). Given the flexibility in the definition of a dietary pattern as the reviewer noted, it is important to form a representative definition of dietary patterns for us to proceed with the downstream analyses of dietary patterns.

We agree with the reviewer that the flexibility in the definition of dietary patterns is important to consider. In the revised manuscript, we have therefore performed an additional analysis of robustness in which we computed amino acid profiles in two additional schemes of Mediterranean diet and two additional schemes of Atkins diet to figure out how the definition of dietary patterns affect the resulting amino acid profiles. Definitions of the two additional Mediterranean diet schemes were developed based on: (1) the nine components of the Mediterranean diet score (MDS) which is a quantitative metric widely used in nutritional epidemiology to assess adherence to the Mediterranean diet (Trichopoulou et al, N Engl J Med 2003, PMID: 12826634); (2) the Mediterranean diet pyramid in an earlier version of the dietary guidelines for adults in Greece (Archives of Hellenic Medicine 1999, 16(5):516-524). Definition of the two additional Atkins diet schemes were developed based on: (1) the "ongoing weight loss" phase of the Atkins diet, in which the upper limit of daily carbohydrate intake is 50 grams instead of 20 grams; (2) the Atkins diet with limited consumption of red meat, in which the consumption of red meat was strictly constrained to zero. We performed one-way ANOVA to compare the amino acid profiles across alternative schemes of the same dietary pattern and compared the resulting F-statistic values to those of the comparison across the original 10 dietary patterns and across three dietary patterns randomly selected from the original 10 dietary patterns (Response Figure 8). We found that the variability of amino acid composition was

indeed lower across the different schemes of Mediterranean diet and Atkins diet compared to the variability across the original 10 diets, hence confirming that the amino acid signatures of human dietary patterns are robust to changes in the definitions of dietary patterns. We have included these results in the revised manuscript.

Line 157-159. "We reconstructed the dietary amino acid intake profiles in more than 30,000 human subjects in the United States based on dietary records in the National Health and Nutrition Examination Survey (NHANES) 2007-2014 datasets." There might be a misunderstanding about dietary assessment methodology in NAHNES. As far as I know, NHANES collected one or two 24-hour dietary recalls rather than dietary records. Dietary records are considered the "gold standard" in dietary assessment in free-living populations but they are extremely burdensome and expensive and thus not typically not feasible. 24-hr recalls are more practical in large surveys like NHANES. The authors need to describe the methodology and its pros and cons more clearly. The authors also need to take advantage of the repeated measures of diets to reduce measurement errors in self-reported amino acid intakes.

We thank the reviewer for the helpful suggestion. In the revised manuscript, we have included additional discussion on the pros and cons of the 24-hour dietary recalls, which was used to collect the dietary data in NHANES. For the comment on using the repeated measures of diets, it is worth noting that we have already done this in the original manuscript. The dietary intake values were computed by averaging over the two 24-hour recalls, which helps reduce noises and measurement errors in the self-reported dietary profiles. We have also included additional discussion to emphasize this point in the revised manuscript.

To assess the consistency of the self-reported dietary intake profiles between the two 24-hour recalls, we have performed a permutation-based analysis in the revised manuscript. Briefly, we computed the Euclidean distance between the two 24-hour dietary intake profiles for each individual, and then randomly perturbed the dietary records so that each dietary intake profile from the first day was paired with a random dietary intake profile taken from the second day but not for the same individual. We then computed the Euclidean distance between each pair of dietary intake profiles and compared the distribution of the Euclidean distances between the

randomly paired dietary records (“Unmatched”) to the distribution of Euclidean distance between the two dietary intake profiles from the same person (“Matched”). We found that the distances between unmatched dietary profiles were significantly greater than the distances between matched dietary profiles from the same individuals (Response Figure 8, Wilcoxon’s rank-sum p -value $< 1e-323$), hence confirming that the self-reported dietary intake profiles, which include the intake of amino acids, were highly consistent between the two 24-hour dietary recalls. The additional results have been included in the revised manuscript.

Line 201-204. "We retrieved the medical records of 18,196 adult subjects in the NHANES 2007-2014 datasets and defined quantitative scores describing the incidences of hypertension, obesity, cancer, and diabetes based on the examination, laboratory, and questionnaire datasets." It is unclear how the authors did this. Did they actually retrieve the medical records of the participants? Did they actually assess incidences of the diseases? The NHANES dataset contains prevalence rather than incidence of these diseases. Given the nature of the national survey, it is unlikely for researchers to retrieve the medical records of the participants.

We appreciate the comment. After carefully studying the distinction between disease incidence and prevalence, we agree with the reviewer that the disease-related information in NHANES and the corresponding health variables we defined in our study reflect prevalence of disease, not incidence. However, although we could not distinguish new and pre-existing disease based on the information in NHANES, they are still useful in identifying associations between disease burden and dietary intakes. We have included discussion of this point in the revised manuscript.

Line 204-207. "We first computed partial Spearman’s rank correlation coefficients as a metric to evaluate the association between dietary amino acid composition and the incidences of the four diseases while controlling for confounders including demographic and lifestyle-related factors." Again, it is unclear whether the authors looked at disease incidence or prevalence. Spearman's rank correlations are not an appropriate methodology to evaluate the associations between amino acids and disease risk because of the complex relationship between diet and disease. Typically, multivariate logistic or Cox proportional hazards models are used to estimate disease associations in epidemiologic studies. It is also important to control for potential confounding factors using the multivariate models. Besides dietary and lifestyle confounders, it is also important to control for health care access and neighborhood SES.

We thank the reviewer for the helpful suggestions. In the revised manuscript, we have made substantial changes to the epidemiological and statistical approaches according to this comment. The changes include: (1) using multivariate logistic regression models that take both dietary variables and potential confounding factors as input variables to model the complex associations between dietary variables and disease prevalence while controlling for confounding factors; (2) including more potential confounding factors in the analysis including access to health care and insurance. We have built a logistic regression model with elastic net regularization that links dietary amino acids, other dietary variables, and all potential confounding factors to health outcomes:

$$p(y = 1 | \mathbf{x}_{AA}, \mathbf{x}_{nut}, \mathbf{x}_c) = \frac{e^{(\mathbf{w}_{AA}^T \mathbf{x}_{AA} + \mathbf{w}_{nut}^T \mathbf{x}_{nut} + \mathbf{w}_c^T \mathbf{x}_c + b)}}{1 + e^{(\mathbf{w}_{AA}^T \mathbf{x}_{AA} + \mathbf{w}_{nut}^T \mathbf{x}_{nut} + \mathbf{w}_c^T \mathbf{x}_c + b)}}$$

In addition to evaluating the predictive power of the model using the area under the receiver operating characteristic (ROC) curve, we also computed feature importance of each variable using standardized regression coefficients to assess its contribution in determining the health outcomes (Response Figure 3). This approach, while offering an unbiased approach to evaluate the effects of dietary intakes on human health while controlling for confounding and interactions between nutrients, further supports our conclusion that dietary intake of amino acids are important determinants of human health outcomes. We found that the prevalence of all four diseases can be predicted from nutritional intakes (AUC>0.6 for all four diseases, Response Figure 9A) and were all affected by dietary amino acid intake (more than 20% of amino acid variables have non-zero regression coefficients for all four diseases, Response Figure 9B). The computed variable importance for amino acids were also comparable to or higher than those for dietary carbohydrate and fat intake (Response Figure 9B and 9C). We have included these results in the revised manuscript.

Line 230-238. "AUC = 0.55 for amino acids compared to 0.55 for macronutrient composition in predicting obesity, and AUC = 0.53 for amino acids compared to 0.52 for macronutrient composition in predicting hypertension." These AUCs are all fairly low and will not be useful in predicting disease outcomes in clinical or public health settings.

We thank the reviewer for raising this concern and agree with the reviewer that there is still space to improve the predictive power of machine learning models used in our study. One possible reason that the AUCs in the previous version of manuscript were low was because the dietary variables we used to train the machine learning models have all been adjusted for confounding factors by fitting a linear regression model linking the confounding factors to the dietary variables and using the residues instead of the actual dietary intakes as inputs of the machine learning models. This approach might underestimate the association between dietary intake of nutrients and disease prevalence.

In the revised manuscript, we have developed multivariate logistic regression models that take all nutrient intakes and potential confounding factors as the input variables, instead of models that only consider the adjusted dietary variables. As we mentioned above, we found that the prevalence of all four diseases can be predicted from nutritional intakes ($AUC > 0.6$ for all four diseases, Response Figure 9A) and were all affected by dietary amino acid intake (more than 20% of amino acid variables have non-zero regression coefficients for all four diseases, Response Figure 9B). The computed variable importance for amino acids were also comparable to or higher than those for dietary carbohydrate and fat intake (Response Figure 9B and 9C). We have included these results in the revised manuscript.

Line 306-309. "Hence, based on such strategy, we developed an AI for designing diets including the Mediterranean, Paleo, and ketogenic diet (Figure 5f). Each diet contains a variety of foods from diverse sources and keeps the features of the corresponding dietary pattern." These food choices seem arbitrary and restrictive for individuals who want to follow certain dietary patterns. There should be more flexibility for food choices, allowing for more diversity in not just amino acids but also other macronutrient compositions across various dietary patterns. Obesity is not the best health outcome for optimizing food patterns because of the problem of reverse causation (i.e. people may change their diet as a result of their weight).

We thank the reviewer for raising this concern and apologize for the confusion here.

The choice of foods (shown in Figure 5f) was made by the AI for diet design, which searches all combinations of foods that satisfy the requirements of a dietary pattern for combinations of foods (in other words, diets) that lie on the Pareto front determined by the two objectives of minimizing the total intake of AAs-to-minimize and maximizing the total intake of AAs-to-maximize. Hence the foods were not arbitrarily selected but rationally designed. We have included clarification of this point in the revised manuscript.

We thank the reviewer for pointing out the issue regarding using obesity as the health outcome and completely agree that the associations identified in our analysis could be the consequences of reverse causation. The reason for using obesity as the health outcome in developing the AI for food design was that obesity was found to have the strongest association with dietary amino acid intake in our previous analysis. In the revised manuscript, we have modified the AI for food design to allow it to also design diets minimizing the prevalence of diabetes. Furthermore,

we have also validated the association between disease outcomes and deviation from Pareto surface with the NHANES 2015-2016 dataset as an external validation set.

Response Figure 10. Relationship between dietary AA intake, obesity, and diabetes. (A) AAs negatively or positively associated with obesity prevalence. (B) Association of obesity prevalence with total intake of AAs-to-minimize (positively associated with obesity) and AAs-to-maximize (negatively associated with obesity). (C) Association between obesity prevalence and deviation from Pareto surface. (D) Chi-squared p-values quantifying the associations shown in (C). (E) AAs negatively or positively associated with diabetes prevalence. (F) Association between diabetes prevalence and deviation from Pareto surface. (G) Chi-squared p-values quantifying the associations shown in (F).

First, we have improved the approach used to identify amino acids positively or negatively associated with disease outcomes by integrating the partial Spearman correlation analysis and logistic regression models: if an amino acid has a positive partial Spearman correlation coefficient with the disease and a positive linear coefficient in the regularized logistic regression model predicting the prevalence of that disease (Response Figure 9), that amino acid was then identified as positively associated with the disease. Similarly, an amino acid was identified as negatively associated with a disease if its partial Spearman correlation coefficient with the disease and linear coefficient in the corresponding logistic regression model were both negative. According to this criterion, the amino acids negatively associated with obesity were phenylalanine, tryptophan, and valine, while amino acids positively associated with obesity were glycine and methionine. Linear combinations of these amino acids were able to predict the prevalence of obesity with higher accuracy than those identified in the previous version of manuscript (Chi-squared p-value = 1.2e-11 compared to 9.0e-8 for AAs positively associated with obesity, and 9.1e-11 compared to 4.9e-10 for AAs negatively associated with obesity, Response Figure 10B). We have then applied this approach to investigate the association between dietary amino acid intake and diabetes, and found that the prevalence of diabetes was significantly associated with the deviation from Pareto front defined by the two objectives of

minimizing total AAs positively associated with diabetes and maximizing total AAs negatively associated with diabetes (Chi-squared p-value < 0.05 for all 10 diets, Response Figure 10).

Response Figure 11.

Associations between disease outcomes and deviation from Pareto surface in an external validation dataset. (A)

Correlation between the actual protein intake and total AA intake in the external validation set. **(B)** Association between obesity prevalence and deviation from Pareto surface in the external validation set. **(C)**

Same as in **(B)** but for diabetes. **(D)** Chi-squared p-values quantifying the associations shown in **(B)** and **(C)**.

To further validate the relationship between dietary amino acid and diseases, we have then performed additional analysis with an external NHANES dataset, the NHANES 2015-2016 dataset, which was not included in the original analysis. We reconstructed dietary amino acid intake for all dietary records included in the NHANES 2015-2016 datasets, and validated the reconstructed dietary AA intake by correlating total daily AA intake to the known values of total daily protein intake (Response Figure 11A). We then correlated the prevalence of diabetes and obesity to the deviation from Pareto fronts in the external dataset, and found that for both diabetes and obesity, the disease prevalence was still significantly positively associated with deviation from the Pareto fronts defined based on AAs positively or negatively associated with the diseases identified based on the original NHANES 2007-2014 datasets (Chi-squared p-value < 0.05 for both diabetes and obesity and all 10 diets, Response Figure 11B-D). The external validation further support our conclusion that health outcomes can be predicted by the deviation from Pareto fronts determined by the two objectives of maximizing and minimizing total intake of certain amino acids. We have included all these additional analyses in the revised manuscript.

Reviewer #3 (Remarks to the Author):

This is an interesting paper showing the variation of amino acids is greater than fat and carbohydrate categories among food and dietary patterns. Also, it studies the association between amino acid intake derived from self-reported data and health outcomes. I have a few comments regarding the model used and the conclusion made in the paper which is summarized by following parts:

We thank the reviewer for the positive remarks and helpful comments. In the revised manuscript, we have performed several additional statistical analyses to address these concerns.

(1) Amino acid landscape of human food

-It is unclear whether the raw abundance data or standardized data are used for the PCA analysis. Given the high % variation explained in PC1, it will be of interest to see the loading factor so the PC can be interpreted.

We thank the reviewer for the helpful suggestion and apologize for the confusion. In the PCA analysis, standardized data in which amino acid levels were normalized to total amino acid levels were used. We have clarified this point in the revised manuscript. We also agree that the loading factors are helpful in interpreting the meaning of the PCs. We have thereby computed loadings of each amino acid in PC1, and found that amino acids with largest contribution to PC1 were glutamate/glutamine, proline, lysine, and aspartate/asparagine (Response Figure 12). Each of these amino acids is enriched in at least one major category of human foods, e.g. enrichment of glutamate/glutamine is a signature of cereals, while lysine is abundant in animal-derived protein. Hence, their high contribution to PC1 suggests that the variation explained by PC1 is dominated by difference in amino acid profile between different types of foods. We have included this result and related discussions in the revised manuscript.

-The unit is amino acid within each gram of food, but it might be interesting to see amino acid within each cal of food given that daily intake will be more related to total energy rather than the total weight of the food.

We thank the reviewer for this comment. In the original manuscript, most analyses of the amino acid levels were performed using amino acid levels normalized to total amino acid levels in foods, diets, or dietary records. The purpose of doing the normalization was to control for both total protein and total energy. We have clarified this point in the revised manuscript.

-plot 1b suggests most amino acid abundance are right-skewed, so it seems the use of ANOVA and PCA shall be after log transformation of the abundance. Or perhaps the relative abundance is less skewed.

We thank the reviewer for the helpful suggestion. We have repeated the ANOVA and PCA analyses with log-transformed amino acid abundances and found that the results were highly consistent with that based on the analysis using raw, non-transformed amino acid abundances, hence validating the robustness of our findings to log transformation of the amino acid profiles (Response Figure 13). We have included these results in the revised manuscript.

Response Figure 13. ANOVA and PCA of the log-transformed amino acid profiles. (A) PCA of log-transformed amino acid profiles in foods. (B) F-statistic from one-way ANOVA comparing log-transformed abundances of amino acids, amino acid subtypes, carbohydrates and fats across foods. (C) Same as (A) but for the analysis of diets. (D) Same as (B) but for the comparison across diets.

(2) Amino acid landscape of human dietary patterns

-Each human dietary pattern is defined by a linear constraint on food intake. However, this might be just a necessary condition for a pattern rather than a sufficient condition. So the range derived from linear programming (no constraint found) can be too wider than it should be in the real population.

-Based on the definition, there could be overlap between different dietary patterns, getting PCA by sampling uniformly from the possible range lacks good interpretation to my opinion as the real population with the certain dietary patterns are less likely to be uniformly distributed and when some value can be within multiple dietary patterns, the correlation shall be considered.

We thank the reviewer for raising this concern. We completely agree that the actual distributions of nutrient and food intake in people adhering to a specific dietary pattern could be much more complicated than the uniform distribution we assumed in our modeling framework. However, precisely modeling the actual probability distribution of nutrient intake in

all people adhering to a specific dietary pattern is extremely challenging, given the difficulty in collecting and accessing reliable dietary data in a cohort that is large enough to represent subpopulations consuming the numerous different diets. Therefore, we chose to use linear constraints to mathematically model different dietary patterns, which can be easily fit into the form of linear programming problems hence allowing efficient sampling of diets consistent with that dietary pattern. Although such approach is unable to completely capture the complexity in the distribution of nutrient intake, it was able to yield interpretable results that can be validated by external studies. For instance, the difference in branched-chain amino acid levels between plant-based diet and ketogenic diet predicted by our model could be validated by the difference in circulating levels of amino acids in individuals consuming the two diets (Figure S2d in the original manuscript).

Response Figure 14. Comparison between model-predicted and actual amino acid signature of ketogenic diet.

In the revised manuscript, we have chosen ketogenic diet as an example to further validate the model-predicted amino acid signatures using the human dietary records included in the NHANES 2007-2014 datasets. We have defined for each individual a ketogenic score quantifying this person's adherence to the ketogenic diet:

$$K = -(\max(f_c - 0.05, 0))^2 - (\min(f_l - 0.7, 0))^2$$

In which f_c is the fraction of calories from dietary intake carbohydrate, and f_l is the fraction of calories from dietary intake of fat. For each amino acid, we then computed the Spearman's rank correlation coefficient between its intake and the ketogenic score. The computed correlation coefficients indicate associations between dietary intake of amino acids and adherence to ketogenic diet: amino acids enriched in ketogenic diets will have positive correlation coefficients, while amino acids with lower intake in ketogenic diet will have negative correlation coefficients. Hence, they were able to serve as indicators of amino acid signatures associated with ketogenic diet. By comparing these amino acid signatures of ketogenic diet directly computed from human dietary records to the amino acid profiles of ketogenic diet predicted by our modeling framework using linear programming (defined as the difference between mean amino acid abundance in computationally sampled ketogenic and other diets), we found that the theoretical predictions were highly consistent with actual signatures of the ketogenic diet (Spearman's correlation = 0.5, p-value = 0.03, Response Figure 14). We believe that such additional analysis can further confirm that our modeling framework was able to

quantitatively evaluate amino acid profiles of different dietary patterns. We have included these results in the revised manuscript.

(3) Amino acid landscape of human dietary records

-Four imputation scenarios are considered and the one with the best 5-fold CV correlation is selected. But it is unclear within each scenario, whether single or multiple imputations are used? It is well known single imputation will not lead to correct statistical inference. More details are needed if multiple imputations are used.

-It is unclear whether and how survey weights are taken into consideration when using NHANES data.

We thank the reviewer for pointing out the importance of using multiple imputations to deal with the missing data and considering survey weight in the statistical analysis.

Regarding the comment on the four imputation scenarios used in this study, these imputation scenarios were formed by applying two different algorithms for data imputation to either transformed (i.e. amino acid levels were normalized to the total amino acid level in a food) or untransformed data (i.e. absolute amino acid levels with the unit gram per gram of food were used). The two algorithms are Multiple Imputation by Chained Equations (implemented in the R package MICE) and random forest regression (implemented in the R package missForest) – both methods are based on multiple imputations of the data. In the revised manuscript, we have included additional discussion on the imputation to clarify this point.

Regarding the comment on considering the survey weights, they were ignored in the previous version of manuscript. The reason for not using survey weights in the analysis of NHANES data was that we were only interested in the association between dietary variables and human health, which we assumed to be constants across different subpopulations. Nevertheless, we agree with the reviewer that considering the sample weights in the analysis will improve the accuracy of estimated associations.

Hence, in the revised manuscript, we have developed a regularized logistic regression model that predicts disease outcomes from the dietary variables and potential confounding factors:

$$p(y = 1 | \mathbf{x}_{AA}, \mathbf{x}_{nut}, \mathbf{x}_c) = \frac{e^{(w_{AA}^T \mathbf{x}_{AA} + w_{nut}^T \mathbf{x}_{nut} + w_c^T \mathbf{x}_c + b)}}{1 + e^{(w_{AA}^T \mathbf{x}_{AA} + w_{nut}^T \mathbf{x}_{nut} + w_c^T \mathbf{x}_c + b)}}$$

This model also takes the survey weights into consideration by integrating the survey weights in the loss function used in training the model. This approach, while offering an unbiased approach to evaluate the effects of dietary intakes on human health while considering the survey weights, further supports our conclusion that dietary intake of amino acids are important determinants of human health outcomes. We found that the prevalence of all four diseases can be predicted from nutritional intakes (AUC>0.6 for all four diseases, Response Figure 15A) and were all affected by dietary amino acid intake (more than 20% of amino acid variables have non-zero regression coefficients for all four diseases, Response Figure 15B). The computed variable importance for amino acids were also comparable to or higher than those for dietary carbohydrate and fat intake (Response Figure 15B and 15C). Hence, when the survey weights have been considered, it did not change the main conclusions of our study. We have included these results in the revised manuscript.

(4) Association of Amino acid and human health

-The self-reported dietary data are known to have systematic bias. The author proposes to use serum Vitamin D and blood urea nitrogen to determine the quality. Given that the food contains vitamin D and amino-acid could be very different, to me only the protein plot in figure 3b is useful and the correlation is very low.

We thank the reviewer for raising this concern. We completely agree with the reviewer that the systematic bias in the self-reported dietary data should be properly assessed. However, the laboratory test data in the NHANES database that can be used to validate the dietary intake is very limited. We do not expect a very strong correlation between dietary protein intake and blood urea nitrogen since blood urea concentration is affected by many factors including kidney function. Moreover, there is considerable overlap between foods rich in vitamin D and foods rich in protein, for instance egg yolk, oily fish and liver, hence the correlation between dietary vitamin D intake and blood level of vitamin D could also support the reliability of intake data for protein-rich foods. This is also supported by the significant positive correlation between the abundance of vitamin D and protein in human foods, and the significant positive correlation between the intake of vitamin D and protein in human dietary records (Response

Figure 16). We have included these additional results and related discussions in the revised manuscript.

-since it was known the total energy and protein are associated with many diseases, the author shall consider relative abundance as exposure or include total energy or total protein intake as an adjustment

We thank the reviewer for pointing out this and apologize for the confusion. Actually, we have already adjusted for total protein intake by normalizing the absolute intakes of single amino acids to the total intake of all amino acids. We have clarified this point in the revised manuscript.

-the AUC shows no difference in the prediction power between amino acid and carbohydrate and fat intake. $\text{AUC} < 0.6$ is generally considered poor, so even statistically significant, $\text{AUC} = 0.53$ and $\text{AUC} = 0.52$ are not really clinical meaningful differences.

-use logistic regression without transformation on skewed dietary variables might face the model-misspecification problems. The goodness of fit tests is recommended.

We thank the reviewer for raising this concern. One possible reason that the AUCs from the original analysis were low was because the dietary variables we used to train the machine learning models have all been adjusted for confounding factors by fitting a linear regression model linking the confounding factors to the dietary variables and using the residues instead of the actual dietary intakes as inputs of the machine learning models. This approach might underestimate the association between dietary intake of nutrients and disease prevalence. In the revised manuscript, we have developed multivariate logistic regression models that take all nutrient intakes and potential confounding factors as the input variables, instead of models that only consider the adjusted dietary variables (Response Figure 15). That model not only incorporated the survey weights in the NHANES datasets, but also improved the predictive

power ($AUC > 0.6$ for all diseases, Response Figure 15A), thus better supporting our findings on dietary amino acids and human health.

To assess whether log-transformation of the independent variables might affect the outcomes of the logistic regression model, we have also performed an additional analysis in which we log-transformed the skewed, nonnegative independent variables used in the model to better mimic a normal distribution and used the log-transformed variables to train the logistic regression model. We then compared the values of AUC and regression coefficients between the models with and without log-transformation of the independent variables, and found that log-transformation of the dependent variable had little impact on the interpretation (i.e. regression coefficients for the independent variables, Response Figure 17A) and performance (i.e. AUC values, Response Figure 17B) of the model. We have included these additional results in the revised manuscript.

Response Figure 17. Comparison between the logistic regression models that use raw or log-transformed independent variables. (A) Scatter plots comparing regression coefficients for all independent variables between the two models. (B) Comparison of the AUC values between the two models.

-not clear whether survey weight are taken into account

We thank the reviewer for raising this concern. As we mentioned above, we have developed a regularized logistic regression model which takes the survey weight into consideration (Response Figure 15).

REVIEWER COMMENTS

Reviewer #1 (Remarks to the Author):

I greatly appreciate the considerable effort taken by the authors to address my comments and those of the other referees (among which there was significant concordance around some key issues). I am happy with the responses and re-analyses.

Reviewer #2 (Remarks to the Author):

The revised manuscript is highly responsive to my earlier comments and has substantially improved. I have several additional questions:

1. The analysis is based on dietary intakes of AAs without considering the digestion and absorption of these AAs and food sources. This needs to be discussed.
2. Dietary intakes of different AAs are highly correlated because some come from food sources. It's unclear how the models take into collinearity of these AAs, especially in analyzing various dietary patterns.
3. The variability of AAs is likely to be correlated with the amount in specific foods or diets. For example, the intake of glycine is quite low in the population, which can lead to overall low variability.
4. The AAs that are positively or negatively correlated with obesity prevalence are not particularly meaningful given these are cross-sectional correlations without adjustment for important confounding factors such as dietary fatty acids, fiber, etc. For example, the study found that tryptophan intake was negatively correlated with obesity. However, many previous longitudinal and intervention studies have shown higher tryptophan and its metabolites are associated with increased insulin resistance and type 2 diabetes. The authors need to be explicit about the exploratory and proof-of-concept nature of their analyses. At this point, the results from the ML analyses have little clinical or public health utility unless they are verified in future longitudinal cohort studies and intervention trials.

Reviewer #3 (Remarks to the Author):

The revision improves the quality of the paper and the new analysis results are slightly more promising with larger AUCs. The author fully addressed most of my concerns except the following ones:

- (1) I agree with the author that it is challenging to model the actual probability distribution of nutrient intake in specific dietary patterns. However, some discussion shall be added to explicit warning readers of the potential pitfall of using the approach of univariate range with uniform distribution derived from this analysis.
- (2) The statistical significance of the association between Vitamin D and protein is not quite relevant here due to the large sample size. The magnitude of the positive correlation shown in response to figure 16 is still weak and the correlation among foods and among self-reported data are quite different. So using one variable to evaluate the quality of the other could be problematic. I don't feel these plots (Supplement figure 6 (d)(e)) are really useful and would recommend removing and just listing such bias in self-reported data as a limitation.

Reviewer #1 (Remarks to the Author):

I greatly appreciate the considerable effort taken by the authors to address my comments and those of the other referees (among which there was significant concordance around some key issues). I am happy with the responses and re-analyses.

We thank the reviewer for the positive comment on our revisions and insightful advice on the previous version of manuscript that have greatly helped us improve our work.

Reviewer #2 (Remarks to the Author):

The revised manuscript is highly responsive to my earlier comments and has substantially improved. I have several additional questions:

We thank the reviewer for the positive remarks and the constructive suggestions. We have further revised the manuscript to address the reviewer's additional questions.

1. The analysis is based on dietary intakes of AAs without considering the digestion and absorption of these AAs and food sources. This needs to be discussed.

We thank the reviewer for the helpful suggestion. The reason that digestion and absorption of amino acids is not considered in our analysis is that these factors are difficult to quantify without physiological data due to the high variability in amino acid metabolism between individuals in a population. We do agree with the reviewer that these factors have important contribution to how dietary amino acids affect human health and have included additional discussion about that point in the revised manuscript.

2. Dietary intakes of different AAs are highly correlated because some come from food sources. It's unclear how the models take into collinearity of these AAs, especially in analyzing various dietary patterns.

We thank the reviewer for raising this concern and apologize for the confusion. In our study, the models of dietary patterns do not explicitly consider the collinearity between amino acids. This is because these dietary patterns are defined based on the choices of foods (e.g. olive oil in Mediterranean diet, vegetables in plant-based diet, and so on) or intake of major macronutrients (e.g. carbohydrate and fat, which is considered in Atkins diet and ketogenic diet) in that dietary pattern. Abundances of amino acids in these dietary patterns were computed by randomly sampling diets satisfying the constraints related to the required consumption of foods or intake of nutrients for that dietary pattern (i.e. the feasible region). The correlation between abundances of amino acids in a dietary pattern is implicitly encoded in these

constraints because it is a direct consequence of the geometry of the feasible region. We have included discussion about this in the revised manuscript.

3. The variability of AAs is likely to be correlated with the amount in specific foods or diets. For example, the intake of glycine is quite low in the population, which can lead to overall low variability.

We thank the reviewer for pointing out the potential relationship between variability and intake of amino acids in human foods and diets. We agree that it is possible that the variability and level of amino acids in human foods and diets are correlated with each other, hence potentially confounding our conclusions on the variability of amino acids in human foods and diets. To exclude this possibility, we have compared the one-way ANOVA F-statistic values to median levels of amino acids in both human foods and dietary patterns (Response Figure 1). We found that, both in human foods and dietary patterns, the level of amino acids and their variability are uncorrelated with each other (Pearson's $R = -0.20$ for foods and 0.02 for dietary patterns). Hence, our conclusions about the variability of amino acids across human foods and dietary patterns are unlikely to be confounded by the absolute level of amino acid intake. We have included this result and related discussion in the revised manuscript.

Response Figure 1. relationship between median level and F-statistic from one-way ANOVA for amino acids in human foods and dietary patterns.

4. The AAs that are positively or negatively correlated with obesity prevalence are not particularly meaningful given these are cross-sectional correlations without adjustment for important confounding factors such as dietary fatty acids, fiber, etc. For example, the study found that tryptophan intake was negatively correlated with obesity. However, many previous longitudinal and intervention studies have shown higher tryptophan and its metabolites are associated with increased insulin resistance and type 2 diabetes. The authors need to be explicit about the exploratory and proof-of-concept nature of their analyses. At this point, the results from the ML analyses

have little clinical or public health utility unless they are verified in future longitudinal cohort studies and intervention trials.

We thank the reviewer for the constructive advice. We agree with the reviewer that our analysis has the limitation of being exploratory and proof-of-concept and have included discussion about this point in the revised manuscript. We have made the attempt to adjust for potential confounding variables in the analysis of correlation between amino acid intake and disease prevalence and the following development of machine learning models, but because the nutrient intakes in human diets are highly correlated with each other, it is difficult to eliminate the confounding effects caused by the coupling between dietary intake of amino acids and that of other nutrients. Regarding the conclusions derived based on our analyses, we completely agree that they need to be verified with future studies such as longitudinal cohorts and intervention trials and have emphasized this point in the revised manuscript.

Reviewer #3 (Remarks to the Author):

The revision improves the quality of the paper and the new analysis results are slightly more promising with larger AUCs. The author fully addressed most of my concerns except the following ones:

We thank the reviewer for the positive comment on our revisions. We have further revised the manuscript following the reviewer's advice.

(1) I agree with the author that it is challenging to model the actual probability distribution of nutrient intake in specific dietary patterns. However, some discussion shall be added to explicit warning readers of the potential pitfall of using the approach of univariate range with uniform distribution derived from this analysis.

We thank the reviewer for pointing out the importance of the distribution of nutrient intake levels in the dietary patterns. As the reviewer correctly notes, in the analysis of amino acid profiles in human dietary patterns, we used a sampling-based approach that applies the hit-and-run algorithm to uniformly sample points in a region determined by a set of linear equality and inequality constraints related to the corresponding dietary pattern. We agree that the marginal distribution of the level of a single amino acid obtained from this approach could be different from the actual distribution of amino acid intake in that dietary pattern and have included discussion about this in the revised manuscript.

(2) The statistical significance of the association between Vitamin D and protein is not quite relevant here due to the large sample size. The magnitude of the positive correlation shown in response to figure 16 is still weak and the correlation among foods and among self-reported data are quite different. So using one variable to evaluate the quality of the other could be problematic. I don't feel these plots

(Supplement figure 6 (d)(e)) are really useful and would recommend removing and just listing such bias in self-reported data as a limitation.

We thank the reviewer for the helpful suggestion. We have removed Supplementary Figure 6d and 6e from the manuscript and discussed this limitation in the revised manuscript.

REVIEWERS' COMMENTS

Reviewer #2 (Remarks to the Author):

The authors did an excellent job in responding to my earlier concerns. I have no additional comments.

REVIEWERS' COMMENTS

Reviewer #2 (Remarks to the Author):

The authors did an excellent job in responding to my earlier concerns. I have no additional comments.

We thank the reviewer for the positive comment and all those constructive suggestions that have greatly helped us improve our manuscript during the previous rounds of revision.